# Deep SE(3)-Equivariant Geometric Reasoning for Precise Placement Tasks

**Ben Eisner** *
Carnegie Mellon University

**Yi Yang, Todor Davchev, Mel Veceric, Jon Scholz**
Google DeepMind

**David Held**
Carnegie Mellon University

## Abstract

Many robot manipulation tasks can be framed as geometric reasoning tasks, where an agent must be able to precisely manipulate an object into a position that satisfies the task from a set of initial conditions. Often, task success is defined based on the relationship between two objects - for instance, hanging a mug on a rack. In such cases, the solution should be equivariant to the initial position of the objects as well as the agent, and invariant to the pose of the camera. This poses a challenge for learning systems which attempt to solve this task by learning directly from high-dimensional demonstrations: the agent must learn to be both equivariant as well as precise, which can be challenging without any inductive biases about the problem. In this work, we propose a method for precise relative pose prediction which is provably SE(3)-equivariant, can be learned from only a few demonstrations, and can generalize across variations in a class of objects. We accomplish this by factoring the problem into learning an SE(3) invariant task-specific representation of the scene and then interpreting this representation with novel geometric reasoning layers which are provably SE(3) equivariant. We demonstrate that our method can yield substantially more precise predictions in simulated placement tasks than previous methods trained with the same amount of data, and can accurately represent relative placement relationships data collected from real-world demonstrations. Supplementary information and videos can be found at this URL.

## 1 Introduction

A critical component of many robotic manipulation tasks is deciding how objects in the scene should move to accomplish the task. Many tasks are based on the relative relationship between a set of objects, sometimes referred to as "relative placement" tasks (Simeonov et al. (2022); Pan et al. (2023); Simeonov et al. (2023); Liu et al. (2022)). For instance, in order for a robot to set a table, the position of the silverware has a desired relationship relative to the plate. These types of problems can be described as geometric reasoning tasks - if you know the locations of each object in the scene, and you know what kind of relationship the objects should have to accomplish the task, then you can logically infer the target position of the objects. We would like our robotic agents to also possess these geometric reasoning faculties to solve such relative placement problems.

In this work, we consider the task of training robotic agents to perform relative placement tasks directly from high-dimensional inputs by watching a small number of demonstrations. Although there have been many recent advances in predicting complex robot behaviors from raw sensor observations (Lee et al. (2020); Miki et al. (2022); Yang et al. (2023); Ha & Song (2022); Akkaya et al. (2019)), high-dimensional observations pose particular challenges in geometric reasoning tasks. For example, the amount by which each object should be moved to reach the goal configuration should be SE(3)-equivariant to the initial locations of the object. However, prior work (Pan et al. (2023)) demonstrates that if a general-purpose neural network is trained on a small number of high-dimensional demonstrations with no additional inductive biases, it will typically not learn to be

---

*Corresponding author: `baeisner@andrew.cmu.edu`

robust to the initial object configurations. Prior work to address this issue (Pan et al. (2023)) incorporates geometric reasoning into the network, but is not provably equivariant. Other work that is provably equivariant (Simeonov et al. (2022)) is outperformed both by (Pan et al. (2023)) and by our method.

To achieve both strong empirical performance and provable equivariance, we propose a visual representation which can be used for equivariant manipulation in relative placement tasks. Our key insight is to decouple representation learning into an invariant representation learning step and an equivariant reasoning step. We achieve this using a novel layer for differentiable multilateration, inspired by work in true-range multilateration (Zhou (2009)). In this work, we propose the following contributions:

- A novel dense representation for relative object placement tasks, which is geometrically interpretable and fully invariant under SE(3) transformations to objects in the scene.

- A method for solving relative placement tasks with differentiable geometric reasoning using a novel layer for differentiable multilateration, which can be trained end-to-end from observations of a small number of demonstrations without additional labels.

- A **provably SE(3)-equivariant** neural network architecture predicting this representation directly from high-dimensional raw inputs.

- A set of simulated experiments which demonstrate superior placement performance in several relative placement tasks - both on precision metrics and on overall success. Our experiments also demonstrate that our method generalizes within a class of objects with reasonable variation, and can be applied to real-world demonstration datasets.

## 2 RELATED WORK

**Object Pose Estimation**: One approach to solving relative placement tasks is via object pose estimation. Recently several approaches have proposed using test-time optimization or correspondence detection to align current observations with demonstration observations with known pose (Florence et al. (2018); Simeonov et al. (2023; 2022)). However, (Simeonov et al. (2022)) requires one of the objects to remain in a fixed location, while (Simeonov et al. (2023)) requires user input to specify relationships of interest - our method has no such restriction and requires no user input. Most similar to our work is TAX-Pose (Pan et al. (2023)), which computes cross-object correspondences to estimate a task-specific alignment between a pair of objects, correcting these correspondences with a learned residual. Both TAX-Pose and our work attempt to learn to predict, for each point on an object, where that point should end up to accomplish the relative placement task. However, we compute this mapping quite differently - while TAX-Pose establishes "cross-correspondence" and updates with a residual vector to regress the next point's location, we learn an SE(3)-invariant representation (which we call RelDist) of the pair of objects, and infer the point's location from this representation using differentiable multilateration. By construction, our framework is fully $SE(3)$-invariant by construction, whereas TAX-Pose is not.

**SE(3) Equivariance in Visual Prediction**: Equivariance of geometric predictions under $SE(3)$ transformations of objects in a scene (including invariance to camera transformations) is a desirable property of many visual prediction systems. Several works design provably equivariant prediction methods (Cohen & Welling (2016); Weiler & Cesa (2019); van der Pol et al. (2020)) and use such methods for robot manipulation (Wang et al. (2022b); Huang et al. (2022); Wang & Walters (2022); Wang et al. (2022a)). Vector Neurons (Deng et al. (2021)) are used to design provably-SE(3) invariant and equivariant versions of standard point cloud analysis architectures like DGCNN (Wang et al. (2019)) and PointNet (Qi et al. (2017)), which we incorporate in the feature-encoder in this work. Neural Descriptor Fields (NDF) (Simeonov et al. (2022)) also use Vector Neurons (Deng et al. (2021)) to achieve provable equivariance, but suffer from suboptimal performance; NDF is outperformed by TAXPose (Pan et al. (2023)) which we compare to in this work. Other methods attempt to learn equivariance via training (Pan et al. (2023)). Another approach is to search over $SO(2)$ or $SO(3)$ (Zeng et al. (2021); Lin et al. (2023)), evaluating candidates with a scoring network to align objects in a scene for a given task. However, such methods have not been demonstrated to perform the precise SE(3) relative placement tasks used in this work.

## 3 BACKGROUND

We consider deep geometric reasoning tasks based on point cloud observations, and build on top of several separate lines of prior work:

**Multilateration:** Given a receiver with an unknown position $p$ in 3D, a set of $K$ beacons with known positions $q_k$ in 3D space, and a set of $K$ measurements $r_k$ of the scalar distance from each beacon to the receiver, the multilateration task is to estimate the position of the receiver $p$. Multilateration is similar to triangulation, except that the direction of the beacons to the receiver is unknown. If the squared error cost is used, this reasoning task reduces to a non-linear least-squares optimization:

$$\min_{p} \quad \sum_{k=1}^{K} \left( ||q_k - p||_2^2 - r_k^2 \right)^2 \tag{1}$$

There are several different approaches to solving this problem, as discussed in a survey paper on multilateration (Sirola (2010)). We leverage the closed-form solution to this problem; see (Zhou (2009)) for details.

**The orthogonal Procrustes problem:** Given two sets of $N$ corresponding 3D points $A$ and $B$, the orthogonal Procrustes problem is to find a rigid transform $T_{AB} \in (R_{AB}, t_{AB})$ which best aligns them. Specifically, if the squared error cost is used, this reasoning task reduces to a constrained least-squares optimization task:

$$\min_{R_{AB}, t_{AB}} \quad \sum_{i=1}^{N} ||R_{AB}A_i + t_{AB} - B_i||_2^2, \qquad \text{s.t.} \quad R_{AB} \in SO(3) \tag{2}$$

This problem has a well-known closed-form solution based on Singular Value Decomposition (SVD) (Sorkine-Hornung & Rabinovich (2017)).

**Equivariance/Invariance:** This work considers two geometric functional properties: $SE(3)$ equivariance (Eq. 3) and invariance (Eq. 4), under transform $T \in SE(3)$, which respectively satisfy:

$$f(x) = y \implies f(Tx) = Ty \tag{3} \qquad\qquad f(x) = y \implies f(Tx) = y \tag{4}$$

**Point Cloud Analysis Architectures**: We make use of Dynamic Graph Convolutional Neural Networks (DGCNN) (Wang & Solomon (2019)) as the primary backbone architecture for learning per-point features. Roughly, each layer of DGCNN performs a two-step operation: 1) it computes a per-point connectivity graph, roughly its neighborhood (in either Euclidean space or latent space), typically using K-Nearest Neighbors; and 2) for each point, performs some sort of learned aggregation of the local neighborhood, which is permutation-invariant. Stacking such layers, an architecture can flexibly capture global and local context of a point cloud.

On top of DGCNN, we also make use of Vector Neurons (Deng et al. (2021)), which is a set of deep learning primitives which are designed to be SE(3)-Equivariant. These primitives can be directly dropped into DGCNN to replace existing layers. Roughly, Vector Neurons accomplish SE(3) equivariance by treating each point's Euclidean coordinate independently (through a batching operation), and constructing primitives that operate on the coordinates independently.

## 4 PROBLEM STATEMENT

**Relative Placement Tasks:** In this work, we consider relative placement tasks, as defined in prior work (Simeonov et al. (2022); Pan et al. (2023); Simeonov et al. (2023)). We borrow mathematical definitions of the task from (Pan et al. (2023)). Relative placement tasks require predicting a rigid body transform which transports a specific rigid object $\mathcal{A}$ from an initial position to a desired, task-specific final position with respect to another rigid object $\mathcal{B}$. Suppose $\mathbf{T}_{\mathcal{A}}^*$ and $\mathbf{T}_{\mathcal{B}}^*$ are $SE(3)$ poses for objects $\mathcal{A}$ and $\mathcal{B}$ respectively such that the placement task is

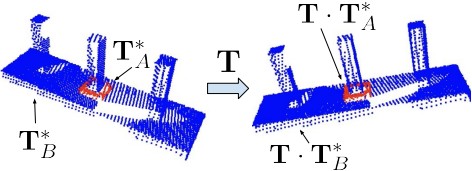

Figure 1: Invariance of relative placement tasks under transformations. In this case, a ring on peg maintains the same relative position under a rigid transformation $\mathbf{T}$.

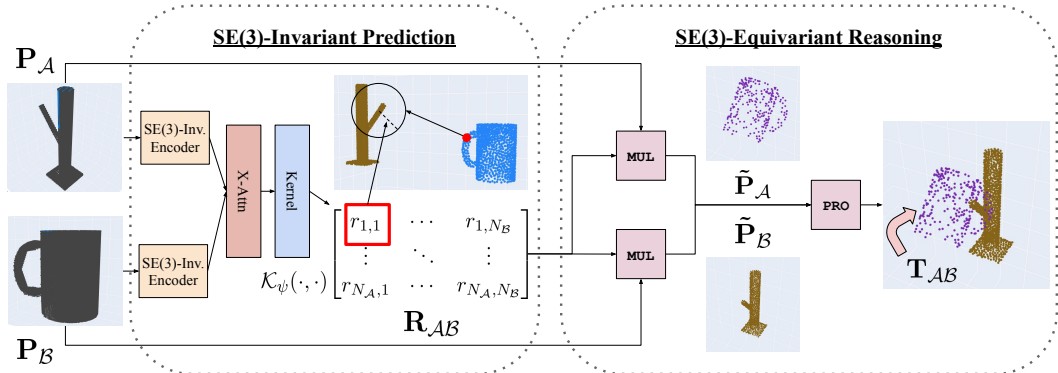

Figure 2: Method overview. First, the point clouds $\mathbf{P}_\mathcal{A}, \mathbf{P}_\mathcal{B}$ are each encoded with a dense $SE(3)$-equivariant encoder, after which cross-attention is applied to yield task-specific dense representations. Then, the kernel matrix $\mathbf{R}_{\mathcal{AB}}$ is constructed through the learned kernel $\mathcal{K}_\psi$. This matrix is then passed into MUL to infer the desired final point clouds, and then passed into PRO to extract a final transform which moves object $\mathcal{A}$ into its goal position.

complete, expressed with respect to some arbitrary world frame. Following Pan et al. (2023), the task is also considered complete if $\mathcal{A}$ and $\mathcal{B}$ are positioned such that, for some $\mathbf{T} \in SE(3)$:

$$\text{RelPlace}(\mathbf{T}_\mathcal{A}, \mathbf{T}_\mathcal{B}) = \textsc{Success} \text{ iff } \exists \mathbf{T} \in SE(3) \text{ s.t. } \mathbf{T}_\mathcal{A} = \mathbf{T} \circ \mathbf{T}_\mathcal{A}^* \text{ and } \mathbf{T}_\mathcal{B} = \mathbf{T} \circ \mathbf{T}_\mathcal{B}^*. \quad (5)$$

**Cross-Pose**: We are interested in predicting a transform which brings object $\mathcal{A}$ to a goal position relative to object $\mathcal{B}$, to satisfy the placement condition defined in Eq. 5. Rather than estimating each object's pose independently and computing a desired relative transform, we would like to learn a function $\mathbf{T}_{\mathcal{AB}} = f(\mathbf{P}_\mathcal{A}, \mathbf{P}_\mathcal{B})$ which takes as input point cloud observations of the two objects, $\mathbf{P}_\mathcal{A}, \mathbf{P}_\mathcal{B}$ and outputs an $SE(3)$ transform which would transport $\mathbf{P}_\mathcal{A}$ into their desired goal position with respect to $\mathcal{B}$. Pan et al. (2023) define this transform as the **cross-pose** for a specific instance of a relative placement task, as follows: when objects $\mathcal{A}$ and $\mathcal{B}$ have been rigidly transformed from their goal configuration by $\mathbf{T}_\alpha$ and $\mathbf{T}_\beta$, respectively, the cross-pose is defined as:

$$f(\mathbf{T}_\alpha \circ \mathbf{T}_\mathcal{A}^*, \mathbf{T}_\beta \circ \mathbf{T}_\mathcal{B}^*) = \mathbf{T}_{\mathcal{AB}} := \mathbf{T}_\beta \circ \mathbf{T}_\alpha^{-1} \quad (6)$$

meaning that applying this transformation to object $\mathcal{A}$ will then transform it back into a goal configuration relative to object $\mathcal{B}$. Note that the cross-pose function is **equivariant** in the sense that $f(T_\mathcal{A} \circ \mathbf{P}_\mathcal{A}, T_\mathcal{B} \circ \mathbf{P}_\mathcal{B}) = T_\mathcal{B} \circ f(\mathbf{P}_\mathcal{A}, \mathbf{P}_\mathcal{B}) \circ T_\mathcal{A}^{-1}$.

Our method assumes that we receive point cloud observations of the scene (e.g. from LiDAR/depth cameras/stereo), and that the pair of objects being manipulated (e.g. the mug and the rack) have been segmented. For training the cross-pose prediction module, we assume access to a set of demonstrations of pairs of objects in their final goal configuration (e.g. demonstrations in which objects $\mathcal{A}$ and $\mathcal{B}$ have poses $\mathbf{T}_\mathcal{A}, \mathbf{T}_\mathcal{B}$ such that $\text{RelPlace}(\mathbf{T}_\mathcal{A}, \mathbf{T}_\mathcal{B}) = \textsc{Success}$, according to Eqn. 6).

## 5 A FRAMEWORK FOR SE(3) EQUIVARIANT MANIPULATION

**Method Overview:** We now describe our method for precise relative object placement which is provably $SE(3)$-equivariant under transformations to individual objects. At a high level, we: 1) Generate SE(3)-invariant per-point features for objects in the scene; 2) Use cross-attention to augment these features with task-specific information; 3) Apply a learned kernel function, which takes in a pair of points on two different objects and predicts how far apart they should be to satisfy the task; 4) Use a differentiable closed-form estimator of the least-squares solution to the multilateration problem to estimate the desired task-specific final location of each point; and 5) Compute the least-squares transform between the estimated point locations and the input locations, using differentiable SVD. See Figure 2 for a high-level overview of our approach.

## 5.1 RELDIST : AN SE(3)-INVARIANT REPRESENTATION FOR RELATIVE PLACEMENT

The key insight of this work for solving relative placement tasks is as follows: suppose that we wish to predict the desired goal position of object $\mathcal{A}$, which should be placed in some location relative to object $\mathcal{B}$. One approach is to predict the desired pose of object $\mathcal{A}$ directly, relative to object $\mathcal{B}$; previous work Pan et al. (2023) shows that this approach has poor performance. Another approach used in prior work that performs better Pan et al. (2023) is to predict the desired location of each point $p_i^{\mathcal{A}}$ in object $\mathcal{A}$. Instead, our approach is to predict a set of desired distance relationships between $p_i^{\mathcal{A}}$ and at least 3 points on object $\mathcal{B}$. Specifically, for each point $p_i^{\mathcal{A}}$ in object $\mathcal{A}$, and for some set of points $\{p_1^{\mathcal{B}}, \ldots, p_K^{\mathcal{B}}\}$ on object $\mathcal{B}$, we predict the desired Euclidean distance $r_{ij} = d(p_i^{\mathcal{A}}, p_j^{\mathcal{B}})$ between points $p_i^{\mathcal{A}}$ and $p_j^{\mathcal{B}}$ in the desired goal configuration. As long as the number of points $K$ used from object $\mathcal{B}$ is at least 3, we can use multilateration (Section 3) to estimate the desired location of $p_i^{\mathcal{A}}$.

Note that these desired distance between point $p_i^{\mathcal{A}}$ on object $\mathcal{A}$ and some point $p_j^{\mathcal{B}}$ on object $\mathcal{B}$ is invariant to the current poses of objects $\mathcal{A}$ and $\mathcal{B}$. Specifically, if object $\mathcal{A}$ is transformed by $\mathbf{T}_\alpha$ and object $\mathcal{B}$ is transformed by $\mathbf{T}_\beta$, then the desired distance between these points is constant:

$$r_{ij} = d(p_i^{\mathcal{A}}, p_j^{\mathcal{B}}) = d(\mathbf{T}_\alpha \circ p_i^{\mathcal{A}}, \mathbf{T}_\beta \circ p_j^{\mathcal{B}}) \tag{7}$$

Further, these distances are scalar values, so they are not defined in any reference frame, which means that they are invariant to shifts in the origin of the coordinate system. We refer to $r_{ij}$ as **RelDist**. This representation gives a convenient target for neural network prediction - one simply needs to predict a set of pairwise invariant relationships $d(p_i^{\mathcal{A}}, p_j^{\mathcal{B}})$ across two objects.

## 5.2 PREDICTING **RELDIST**

We now describe a provably SE(3)-invariant architecture for predicting RelDist :

**Dense SE(3)-Invariant Features:** We assume that we are given a point cloud for objects $\mathcal{A}$ and $\mathcal{B}$, given by $\mathbf{P}_{\mathcal{A}} \in \mathbb{R}^{N_{\mathcal{A}} \times 3}$ and $\mathbf{P}_{\mathcal{B}} \in \mathbb{R}^{N_{\mathcal{B}} \times 3}$, respectively. The first step of our approach is to compute an invariant feature for each point in these point clouds. In other words, the feature should not change if either object is transformed by an SE(3) transformation. Further, we wish each point to have a unique feature. Given point clouds $\mathbf{P}_{\mathcal{A}}$ and $\mathbf{P}_{\mathcal{B}}$, we can use a provably $SE(3)$-Invariant point cloud neural network such as DGCNN Wang & Solomon (2019) implemented using Vector Neuron layers Deng et al. (2021). Alternatively, one could train a standard DGCNN to produce $SE(3)$-Invariant representations, although the resulting features would not be guaranteed to be invariant. We consider both approaches in our experiments.

Specifically, we learn functions $f_{\theta_{\mathcal{A}}}$ and $f_{\theta_{\mathcal{B}}}$ which map the point clouds $\mathbf{P}_{\mathcal{A}}$ and $\mathbf{P}_{\mathcal{B}}$ to a set of invariant $d$-dimensional per-point features $\Phi_{\mathcal{A}}$ and $\Phi_{\mathcal{B}}$:

$$\Phi_{\mathcal{A}} = f_{\theta_{\mathcal{A}}}(\mathbf{P}_{\mathcal{A}}) \in \mathbb{R}^{N_{\mathcal{A}} \times d}, \quad \Phi_{\mathcal{B}} = f_{\theta_{\mathcal{B}}}(\mathbf{P}_{\mathcal{B}}) \in \mathbb{R}^{N_{\mathcal{B}} \times d} \tag{8}$$

**Task-Specific Cross-Attention:** Although the representations $\Phi_{\mathcal{A}}, \Phi_{\mathcal{B}}$ are object specific, the relative placement task requires reasoning about cross-object relationships. For instance, for hanging a mug on a rack, the network needs to reason about the relative location of the mug handle and the peg on the rack. To reason about these cross-object relationships, we learn a cross-attention module, using multi-head cross attention Vaswani et al. (2017), which we sum with the pre-attention embeddings from Eq. 8:

$$\Phi_{\mathcal{A}}', \Phi_{\mathcal{B}}' = \text{X-Attn}_\gamma(\Phi_{\mathcal{A}}, \Phi_{\mathcal{B}}) \tag{9}$$

$$\hat{\Phi}_{\mathcal{A}} = \Phi_{\mathcal{A}} + \Phi_{\mathcal{A}}', \quad \hat{\Phi}_{\mathcal{B}} = \Phi_{\mathcal{B}} + \Phi_{\mathcal{B}}' \tag{10}$$

**A Kernel Function for predicting RelDist :** Let $\phi_i^{\mathcal{A}}$ and $\phi_j^{\mathcal{B}}$ represent the learned embeddings for point $p_i^{\mathcal{A}}$ on object $\mathcal{A}$ and point $p_j^{\mathcal{B}}$ on object $\mathcal{B}$, respectively. We now wish to define a symmetric function $\mathcal{K}$ which takes a pair of embeddings $\phi_i^{\mathcal{A}}$, $\phi_j^{\mathcal{B}}$ and computes a positive scalar $r_{ij} \in \mathbb{R}^+$ which predicts the desired distance between points $p_i^{\mathcal{A}}$ and $p_j^{\mathcal{B}}$ when objects A and B are placed in the goal configuration: $r_{ij} = \mathcal{K}(\phi_i^{\mathcal{A}}, \phi_j^{\mathcal{B}})$. This function $\mathcal{K}$ can be thought of as a type of kernel, for

which there are various choices of functions. In this work, we select a simple learned kernel with the following form:

$$\mathcal{K}_\psi(\phi_i^\mathcal{A}, \phi_j^\mathcal{B}) = \text{softplus}\left(\frac{1}{2}\big(\text{MLP}_\psi(\phi_i^\mathcal{A}, \phi_j^\mathcal{B}) + \text{MLP}_\psi(\phi_j^\mathcal{B}, \phi_i^\mathcal{A})\big)\right) \tag{11}$$

where $\psi$ are a set of learned parameters. Details can be found in the Appendix C.1. This allows us to build the kernel matrix $\mathbf{R}_{\mathcal{AB}}$:

$$\mathbf{R}_{\mathcal{AB}} = \begin{bmatrix} \mathcal{K}_\psi(\phi_1^\mathcal{A}, \phi_1^\mathcal{B}) & \cdots & \mathcal{K}_\psi(\phi_1^\mathcal{A}, \phi_{N_\mathcal{B}}^\mathcal{B}) \\ \vdots & \ddots & \vdots \\ \mathcal{K}_\psi(\phi_{N_\mathcal{A}}^\mathcal{A}, \phi_1^\mathcal{B}) & \cdots & \mathcal{K}_\psi(\phi_{N_\mathcal{A}}^\mathcal{A}, \phi_{N_\mathcal{B}}^\mathcal{B}) \end{bmatrix} \in (\mathbb{R}^+)^{(N_\mathcal{A} \times N_\mathcal{B})} \tag{12}$$

which can be interpreted exactly as the set of RelDist values for all pairs of points $p_i^\mathcal{A}$ and $p_j^\mathcal{B}$ between objects $\mathcal{A}$ and $\mathcal{B}$. Note again that this prediction is $SE(3)$-Invariant for any individual transformations of objects $\mathcal{A}$ or $\mathcal{B}$.

## 5.3 FROM RELDIST TO CROSS-POSE

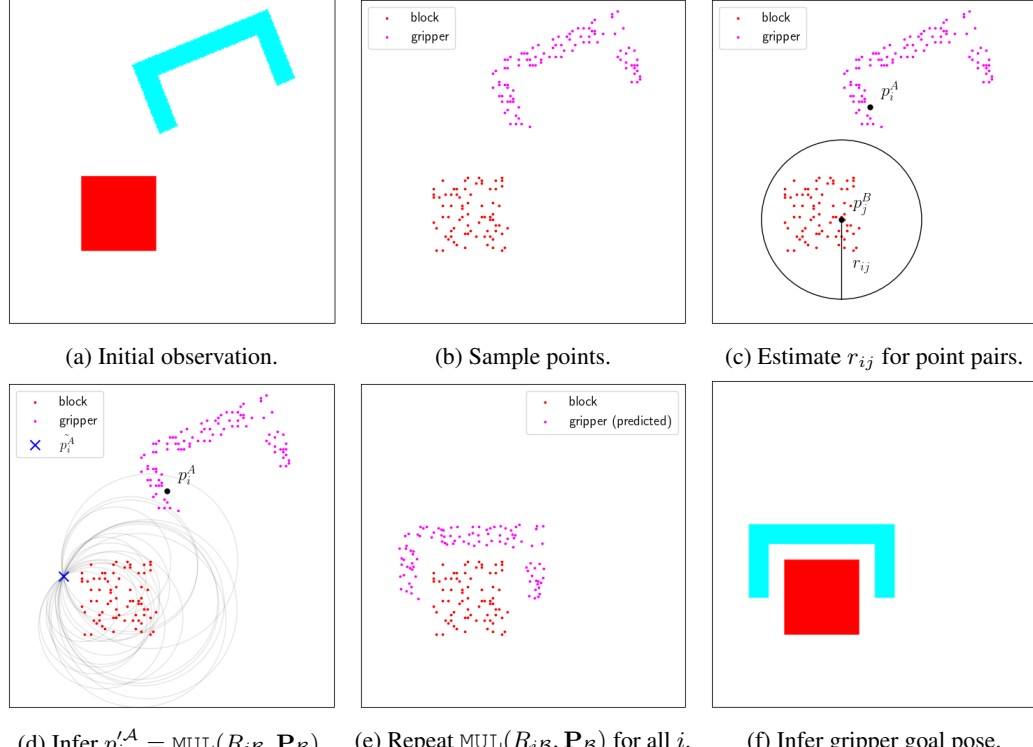

(a) Initial observation.  (b) Sample points.  (c) Estimate $r_{ij}$ for point pairs.

(d) Infer $p_i'^\mathcal{A} = \text{MUL}(R_{i\mathcal{B}}, \mathbf{P}_\mathcal{B})$. (e) Repeat $\text{MUL}(R_{i\mathcal{B}}, \mathbf{P}_\mathcal{B})$ for all $i$. (f) Infer gripper goal pose.

Figure 3: Reasoning with multilateration (a) in a 2D environment with block and gripper. (b) For each sampled point on the gripper, (c) we estimate the desired distances between the gripper point and points on the block. We then use multilateration (d) to extract a least-squares solution to compute the desired gripper point location. Doing this for every point on the gripper, (e) we can reconstruct the desired position for each gripper point. (f) These corresponding points can be used to infer a rigid transform which brings the gripper to the final goal position.

We now describe how we can turn our invariant representation into a Cross-Pose prediction with the desired properties above. A visual walkthrough can be found in Figure 3.

**Estimating Corresponding Points with Differentiable Multilateration:** Given the estimated set of RelDist values $\mathbf{R}_{\mathcal{AB}}$, we would like to estimate the intended position of each point in object $\mathcal{A}$

with respect to object $\mathcal{B}$ (or vice versa). The $i$th row $R_{i\mathcal{B}}$ of $\mathbf{R}_{\mathcal{AB}}$ contains a set of scalars indicating the desired distances from points on object $\mathcal{B}$ to the point $p_i^{\mathcal{A}}$ on object $\mathcal{A}$ when placed in the goal configuration. We would like to use these relative distances to estimate the goal position of this point. Since $\mathbf{R}_{\mathcal{AB}}$ is a predicted quantity and may have some noise, we wish to find the point $p_i'^{\mathcal{A}}$ which minimizes the Mean-Squared error with respect to that row:

$$p_i'^{\mathcal{A}} = \underset{p_i'^{\mathcal{A}}}{\arg\min} \quad \sum_{j=1}^{N_{\mathcal{B}}} \left\| \|p_j^{\mathcal{B}} - p_i'^{\mathcal{A}}\|_2^2 - r_{ij}^2 \right\|_2^2 \tag{13}$$

This describes a nonlinear least-squares optimization problem, a class of problems which has no general closed-form global minimizer. However, it has been shown that there is a differentiable, closed-form global minimizer Zhou (2009) for this specific problem in Equation 13, which we will refer to as the function MUL: $p_i'^{\mathcal{A}} = \texttt{MUL}(R_{i\mathcal{B}}, \mathbf{P}_{\mathcal{B}})$. See Figures 3c and 3d for a visualization of this point estimation process, and Appendix A.1 for details. By computing this function on each row of $\mathbf{R}_{\mathcal{AB}}$, we can compute the desired goal position for each point on $\mathcal{A}$, as: $\tilde{\mathbf{P}}_{\mathcal{A}} = \{\texttt{MUL}(R_{i\mathcal{B}}, \mathbf{P}_{\mathcal{B}})\}_{i \in [N_{\mathcal{A}}]}$ (see Figure 3e). Importantly, this solution to the multilateration problem is provably SE(3)-equivariant to transformations on object $\mathcal{B}$, given a fixed $\mathbf{R}_{\mathcal{AB}}$ (see Appendix B for proof).

**Estimating Cross-Pose with SVD:** Given a set of initial points $P_A$ and their corresponding estimated final task-specific goal positions $\tilde{P}_A$, we can set up a classic weighted Procrustes problem to compute the estimated cross-pose $\mathbf{T}_{\mathcal{AB}}$ between objects $\mathcal{A}$ and $\mathcal{B}$, as defined in Section 3. This has a known closed-form solution based on the Singular Value Decomposition (SVD) (see Appendix A.2) which can be implemented to be differentiable Papadopoulo & Lourakis (2000); Pan et al. (2023). For the relative placement task, once we estimate $\mathbf{T}_{\mathcal{AB}}$, we use motion planning to move object $\mathcal{A}$ by $\mathbf{T}_{\mathcal{AB}}$ into the goal pose. Because MUL and PRO are all differentiable functions, the whole method can be trained end-to-end, using the same loss functions as defined in Pan et al. (2023) (see Appendix C for details).

## 6 Experiments

### 6.1 RLBench Tasks - Achieving Precise Placement

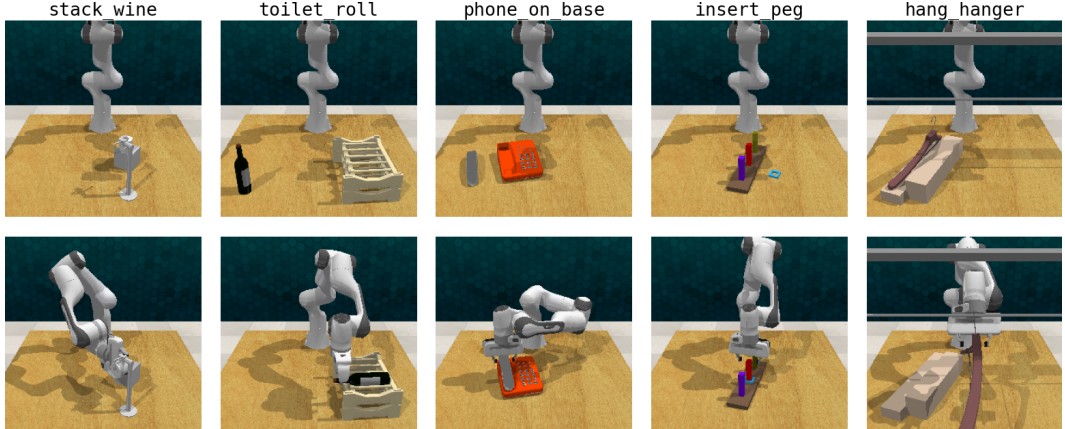

Figure 4: RLBench (James et al. (2020)) relative placement tasks. Top: the initial state of a demonstration. Bottom: the final state of a demonstration, where a successful placement has been achieved.

**Experiment Setup**: To assess the ability of our method to make precise predictions in relative placement tasks, we select 5 manipulation tasks tasks from the RLBench benchmark (James et al. (2020) which require varying amounts of precision to successfully accomplish each task. Specifically, we choose the following tasks: `stack_wine`, `toilet_roll`, `phone_on_base`, `insert_peg`, and `hang_hanger`. Each of these tasks involves placing a singular object in a task-specific configuration relative to some other object. We collect 10 expert demonstrations for each task, and extract

point clouds for initial and final keyframes using ground truth segmentation from multiple simulated RGB-D images. This forms the basis for a per-task relative placement task, where the goal is to predict the desired goal configuration for each object given the initial point cloud. To evaluate the performance of a prediction system, we generate 1000 unseen initial starting positions for the object in the scene, and measure the both the rotation error $\epsilon_r$ and the centroid translation error $\epsilon_p$ of the predictions compared to the goal poses reached in expert demonstrations. We evaluate both TAX-Pose Pan et al. (2023) and our method on each task. Results are presented in Table 1.

| **RLBench Placement Tasks** | | | | | | | | | | |
|---|---|---|---|---|---|---|---|---|---|---|
| | stack_wine | | toilet_roll | | hang_hanger | | phone_on_base | | insert_peg | |
| Method | $\epsilon_r$ (°) | $\epsilon_p$ (m) | $\epsilon_r$ (°) | $\epsilon_p$ (m) | $\epsilon_r$ (°) | $\epsilon_p$ (m) | $\epsilon_r$ (°) | $\epsilon_p$ (m) | $\epsilon_r$ (°) | $\epsilon_p$ (m) |
| TAX-Pose | 1.485 | 0.003 | 1.173 | **0.001** | 5.471 | 0.012 | 4.144 | 0.005 | 7.098 | 0.004 |
| Ours | **0.764** | **0.001** | **1.150** | **0.001** | **0.624** | **0.002** | **0.804** | **0.001** | **1.209** | **0.003** |

Table 1: RLBench Placement Tasks, prediction error (↓ is better): We measure the precision of our method when predicting the cross-pose which brings each object into the goal position. We report both the angular error $\epsilon_r$ (°)and the translational error $\epsilon_p$ (m), compared to the goal poses achieved by expert demonstrations.

**Analysis**: For every task in our evaluation, our method achieves highly precise placements, with rotational errors less than 1.25°and translational errors less than 3 millimeters (and frequently only 1mm). For each task, this level of precision is well within the tolerances for successfully completing the task. Additionally, we substantially outperform our closest baseline (TAX-Pose) in nearly every metric (which matches our translational precision in only on toilet_roll). In particular, our method is 2-9x more precise in rotation on 4 of the tasks. Because each task includes only a single object which varies only in initial pose, this evaluation can be interpreted roughly as a ceiling for how precisely these relative relationships can be represented. We believe that this is strong evidence that our invariant-equivariant architecture is capable of representing significantly more precise relationships in general than prior work. Furthermore, we show in Appendix D that our method can learn highly precise relationships from only a single demonstration.

## 6.2 NDF Tasks - Category-Level Generalization

**Experiment Setup**: To evaluate the ability of our method to generalize precise placements across a class of objects with reasonable variation, we evaluate our method on the NDF relative placement tasks, as proposed in Simeonov et al. (2022). These tasks have the following structure: an object is positioned in a tabletop scene, along with a robot arm with parallel-jaw gripper. The task is to predict two poses for the robot to execute: first, a **Grasp** pose which will lead to a stable grasp on the object when the gripper is actuated; second, a **Place** pose, where the grasped object should be placed to accomplish the task. At training time, the agent is given 10 demonstrations of each stage of the task (grasp and place), with a variety of objects from the same object category. The agent must then learn a relative pose prediction function for each scenario which can generalize to novel instances of objects. There are two variants of each task evaluation: when the object is initialized to be resting upright on the table, bottom-side down ("Upright"), and when the object is initialized at an arbitrary orientation above the table for the robot to grasp ("Arbitrary"). At test time, we randomly sample 100 initial configurations from each distribution and report average metrics across all trials. We report success rates at various thresholds of penetration between the placed object and the scene as a proxy for precision. See Appendix E for more details and motivation for this metric.

We compare our method to the following baselines: Dense Object Nets (Florence et al. (2018)), Neural Descriptor Fields (Simeonov et al. (2022)), and TAX-Pose (Pan et al. (2023)) (see Appendix E.2 for details).

**Analysis**: Our numerical results are presented in Table 3. Our method achieves substantially more precise placements in both the upright and arbitrary mug hanging tasks: while other methods perform well when geometric feasibility is not considered in the evaluation (infinity column), our method yields substantially more feasible predictions in both settings, when penalizing for collisions (penetration thresholds of 1 or 3 cm). Results for the remaining NDF tasks can be found in

| Mug Hanging (Upright/Arbitrary) | | | | | | | |
|---|---|---|---|---|---|---|---|
| | **Grasp** | **Place** | | | **Overall** | | |
| Method | | <1cm | <3cm | ∞ | <1cm | <3cm | ∞ |
| DON[1] | 0.91/0.35 | - | - | 0.50/0.45 | - | - | 0.45/0.17 |
| NDF [2] | 0.96/0.78 | - | - | 0.92/0.75 | - | - | 0.88/**0.58** |
| TAX-Pose [3] | **1.00**/0.50 | 0.15/0.32 | 0.70/**0.75** | 0.93/0.85 | 0.15/0.14 | 0.70/**0.40** | 0.93/0.44 |
| Ours | **1.00**/0.43 | **0.35/0.36** | **0.78**/0.66 | **0.99/0.88** | **0.35/0.19** | **0.78**/0.32 | **0.99**/0.41 |

Table 2: NDF Mug Hanging, Success Rate (↑ is better): The success rate of each method when running the relative placement tasks with the objects starting either upright on the table (left number) or in arbitrary poses above the table (right number). Each column group shows the success rates for the **Grasp** phase and **Place** phase, as well as **Overall** performance (when both grasp and place are successful in a trial). Additionally, for **Place** and **Overall** we report success when maximum penetration is thresholded at the given distance (1cm, 3cm, ∞).

Appendix E; however, our results on the Bottle and Bowl tasks are inconclusive, as implementations of the symmetry-breaking techniques proposed by Pan et al. (2023) - which are crucial to good performance for our closest baseline, TAX-Pose - were not released, and we were not able to reproduce the performance reported by Pan et al. (2023). We also performed multiple ablations to understand the effect of different design decisions of our method; see Appendix F for details.

## 6.3 REAL-WORLD SENSOR EXPERIMENTS

**Experiment Description**: To demonstrate the ability of our method to generalize to real-world sensors, we design an experiment that closely follows the real-world experiments proposed in Pan et al. (2023). Specifically, we choose the real-world mug-hanging experiment from this work. The authors from Pan et al. (2023) have provided us with a dataset real-world mug-hanging demonstrations, collected in the setup displayed in Figure 9. We then train our method in the same way as we did simulated mugs. See Figure 10a and 11 and Table 7 in the Appendix for qualitative and quantitative evaluations of this offline dataset. We find that our method produces predictions consistent with successful mug-hanging.

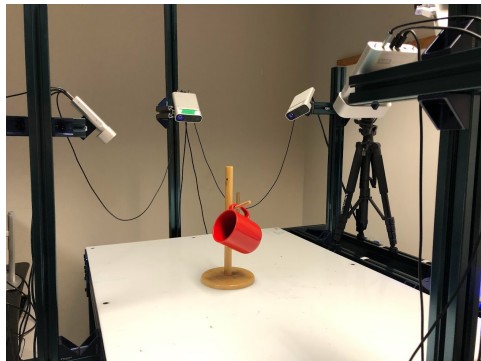

Figure 5: A real-world demonstration of mug-hanging.

## 7 CONCLUSION & LIMITATIONS

In this work, we present a provably $SE(3)$-Equivariant system for predicting task-specific object poses for relative placement tasks. We propose a novel cross-object representation, RelDist, which is an $SE(3)$-Invariant geometric representation for cross-object relationships. We also propose a pair of $SE(3)$-Equivariant geometric reasoning layers that use multilateration and SVD, respectively, to extract a relative pose prediction. Finally, we demonstrate that this representation provides superior performance in high-precision tasks in simulated environments, and is applicable to point cloud data collected in real-world manipulation demonstrations.

This method has several limitations. First, it cannot handle symmetric objects or multimodal placement tasks without an external mechanism for symmetry-breaking, as only a single pose is predicted even when multiple poses may be valid. This might be addressed by exploring generative versions of this model. Further, our method assumes that we have segmented the two task-relevant objects (e.g. the mug and the rack). Still, we hope that our method provides a foundation for future work on SE(3)-Equivariant learning for relative placement tasks.

ACKNOWLEDGMENTS

This material is based upon work supported by the National Science Foundation under Grant No. DGE-1745016 (NSF GRFP), as well as by NIST under Grant No. 70NANB23H178.

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

# Appendix

## A  SE(3)-EQUIVARIANT GEOMETRIC REASONING

In the following sections, we provide the closed-form solutions used for the multilateration and SVD layers, respectively.

### A.1  MULTILATERATION

In Zhou (2009), the problem of multilateration (i.e. estimating the position of a receiver $p$ given a set of distances $r_k$ to $k$ beacons with known positions $q_k$) is considered:

$$\min_{\mathbf{p}^*} \quad \sum_{i=1}^{N} \left| \left| ||\mathbf{p}_i - \mathbf{p}^2||_2^2 - r_i^2 \right| \right|_2^2 \tag{14}$$

Zhou (2009) derives two closed-form, differentiable minimizers to this problem, one of which is for the overspecified case (more than 3 non-co-planar beacons) and for the degenerate case where there are many co-planar beacons. In our approach, we utilize the first solution, as it is simpler to implement and the point distributions for objects are almost never coplanar in our experiments. See Algorithm 1 for pseudocode of this approach. This function is called as $\text{MUL}(\mathbf{R}, \mathbf{P})$.

---

**Algorithm 1** Multilateration (MUL)

---

**Input:** $\mathbf{R} \in \{r_i | r_i \in \mathbb{R}^+\}_{[N]}, \mathbf{P} \in \{\mathbf{p}_i | \mathbf{p}_i \in \mathbb{R}^3\}_{[N]}$
$\quad \mathbf{a} \leftarrow \frac{1}{N} \sum_{i=1}^{N} (\mathbf{p}_i \mathbf{p}_i^T \mathbf{p}_i - r_i^2 \mathbf{p}_i)$
$\quad \mathbf{B} \leftarrow \frac{1}{N} \sum_{i=1}^{N} \left[ -2\mathbf{p}_i \mathbf{p}_i^T + (\mathbf{p}_i^T \mathbf{p}_i)\mathbf{I} + r_i^2 \mathbf{I} \right]$
$\quad \mathbf{c} \leftarrow \frac{1}{N} \sum_{i=1}^{N} \mathbf{p}_i$
$\quad \mathbf{f} = \mathbf{a} + \mathbf{B}\mathbf{c} + 2\mathbf{c}\mathbf{c}^T\mathbf{c}$
$\quad \mathbf{H} = -\frac{2}{N} \sum_{i=1}^{N} \mathbf{p}_i \mathbf{p}_i^T + 2\mathbf{c}\mathbf{c}^T$
$\quad \mathbf{q} = -\mathbf{H}^{-1}\mathbf{f}$
$\quad \mathbf{p}^* = \mathbf{q} + \mathbf{c}$
**Output:** $\mathbf{p}^*$

---

As each operation in the algorithm is differentiable with respect to $\mathbf{R}$, the entire function MUL is differentiable.

### A.2  PROCRUSTES

The problem of finding the (weighted) least-squares minimizing rigid transformation to align two sets of points is the well-known Procrustes problem Sorkine-Hornung & Rabinovich (2017):

$$\min_{R,t} \quad \sum_{i=1}^{N} w_i \, ||R\mathbf{p}_i + t - \mathbf{q}_i||_2^2, \qquad \text{s.t.} \quad R \in SO(3) \tag{15}$$

This problem has a well-known closed-form minimizer, given by the pseudocode in Algorithm 2. Because this algorithm solves the Procrustes problem, we refer to it as PRO for short. This function is called as $\text{PRO}(\mathbf{P}, \mathbf{Q}, \mathbf{w})$ where $\mathbf{w}$ are an optional set of weights and may be omitted if the weights are all equal to 1.

---

**Algorithm 2** Least-squares solution to the Procrustes problem (PRO)

---

**Input:** $\mathbf{P} \in \{\mathbf{p}_i | \mathbf{p}_i \in \mathbb{R}^3\}_{[N]}, \mathbf{Q} \in \{\mathbf{q}_i | \mathbf{q}_i \in \mathbb{R}^3\}_{[N]}, \mathbf{w} \in \{w_i | w_i \in \mathbb{R}^+\}_{[N]}$

$\bar{\mathbf{p}} \leftarrow \frac{\sum_{i=1}^N w_i \mathbf{p_i}}{\sum_{i=1}^N w_i}$

$\bar{\mathbf{q}} \leftarrow \frac{\sum_{i=1}^N w_i \mathbf{q_i}}{\sum_{i=1}^N w_i}$

$\mathbf{X} \leftarrow \{\mathbf{p}_i - \bar{\mathbf{p}}\}_{[N]}$

$\mathbf{Y} \leftarrow \{\mathbf{q}_i - \bar{\mathbf{q}}\}_{[N]}$

$\mathbf{W} \leftarrow \text{diag}(\mathbf{w})$

$\mathbf{S} \leftarrow \mathbf{X}\mathbf{W}\mathbf{Y}^T$

$(\mathbf{U}, \mathbf{\Sigma}, \mathbf{V}^T) \leftarrow \text{SVD}(\mathbf{S})$ ▷ Singular-Value Decomposition

$\mathbf{D} \leftarrow \mathbf{I}$

$\mathbf{D}[N, N] \leftarrow \det(\mathbf{V}\mathbf{U}^T)$

$R \leftarrow \mathbf{V}\mathbf{D}\mathbf{U}^T$

$t \leftarrow \bar{\mathbf{q}} - R\bar{\mathbf{p}}$

**Output:** $R, t$

---

All operations in Algorithm 2 are differentiable (including SVD, if a differentiable implementation is used Papadopoulo & Lourakis (2000); Pan et al. (2023)); therefore PRO is also differentiable.

## B    PROOF OF FULL-SYSTEM EQUIVARIANCE IN PLACEMENT TASKS

We now offer a proof that in relative placement tasks for objects $\mathcal{A}$ and $\mathcal{B}$, our entire proposed system is equivariant under $SE(3)$ transforms to either $\mathcal{A}$ or $\mathcal{B}$. As a reminder, $SE(3)$ equivariance (Eq. 3) and invariance (Eq. 4), under transform $T \in SE(3)$, are defined as follows:

$$f(x) = y \implies f(Tx) = Ty \qquad (16) \qquad\qquad f(x) = y \implies f(Tx) = y \qquad (17)$$

Let us now define what it means for the cross-pose function $\mathbf{T}_{\mathcal{AB}} = f(\mathbf{P}_{\mathcal{A}}, \mathbf{P}_{\mathcal{B}})$ to be equivariant. Recall that we are focusing on relative placement tasks, in which the goal is to move objects $\mathcal{A}$ and $\mathcal{B}$ into a desired relative transformation (defined formally in Section 3). The cross-pose $\mathbf{T}_{\mathcal{AB}}$ defines how object $\mathcal{A}$ needs to be transformed to move into the goal pose relative to object $\mathcal{B}$. For example, suppose $\mathbf{T}_{\mathcal{A}}^*$ and $\mathbf{T}_{\mathcal{B}}^*$ are $SE(3)$ poses for objects $\mathcal{A}$ and $\mathcal{B}$ respectively such that the placement task is complete. Now suppose that objects $\mathcal{A}$ and $\mathcal{B}$ have been rigidly transformed by $\mathbf{T}_{\alpha}$ and $\mathbf{T}_{\beta}$, respectively. Then the cross-pose is given by

$$f(\mathbf{T}_{\alpha} \circ \mathbf{T}_{\mathcal{A}}^*, \mathbf{T}_{\beta} \circ \mathbf{T}_{\mathcal{B}}^*) = \mathbf{T}_{\mathcal{AB}} := \mathbf{T}_{\beta} \circ \mathbf{T}_{\alpha}^{-1} \qquad (18)$$

which is the transform one would need to apply to object $\mathcal{A}$ to move it back into the goal pose, which is defined relative to object $\mathcal{B}$. If we futher transform object $\mathcal{A}$ by $T_{\mathcal{A}}$ and object $\mathcal{B}$ by $T_{\mathcal{B}}$, then the cross-pose becomes

$$f(T_{\mathcal{A}} \circ \mathbf{T}_{\alpha} \circ \mathbf{T}_{\mathcal{A}}^*, T_{\mathcal{B}} \circ \mathbf{T}_{\beta} \circ \mathbf{T}_{\mathcal{B}}^*) = (T_{\mathcal{B}} \circ \mathbf{T}_{\beta}) \circ (T_{\mathcal{A}} \circ \mathbf{T}_{\alpha})^{-1} \qquad (19)$$

$$= T_{\mathcal{B}} \circ (\mathbf{T}_{\beta} \circ \mathbf{T}_{\alpha}^{-1}) \circ T_{\mathcal{A}}^{-1} \qquad (20)$$

$$= T_{\mathcal{B}} \circ \mathbf{T}_{\mathcal{AB}} \circ T_{\mathcal{A}}^{-1} \qquad (21)$$

Thus, we will define a cross-pose function $f$ as equivariant if it satisfies

$$f(T_{\mathcal{A}} \circ \mathbf{P}_{\mathcal{A}}, T_{\mathcal{B}} \circ \mathbf{P}_{\mathcal{B}}) = T_{\mathcal{B}} \circ f(\mathbf{P}_{\mathcal{A}}, \mathbf{P}_{\mathcal{B}}) \circ T_{\mathcal{A}}^{-1} \qquad (22)$$

We can now prove our theorem:

**Theorem 1.** *Let $f$ be the method defined in Section 5, given by*

$$\mathbf{T}_{\mathcal{AB}} = f(\mathbf{P}_{\mathcal{A}}, \mathbf{P}_{\mathcal{B}}) = PRO(\mathbf{P}_{\mathcal{A}}, MUL(\mathbf{R}_{\mathcal{AB}}, \mathbf{P}_{\mathcal{B}})) \qquad (23)$$

*in which $\mathbf{R}_{\mathcal{AB}}$ is computed from $\mathbf{P}_{\mathcal{A}}$ and $\mathbf{P}_{\mathcal{B}}$ using Equations 8, 9, 10, 11, 12; MUL is described in Sec. 5.3 and defined formally in Appendix A.1, and PRO is described in Sec. 5.3 and defined formally in Appendix A.2. Then $f$ is equivariant under $SE(3)$ transformations to either of its inputs $\mathbf{P}_{\mathcal{A}}$ or $\mathbf{P}_{\mathcal{B}}$.*

*Proof.* Assuming:

- $f_{\theta_{\mathcal{A}}}, f_{\theta_{\mathcal{B}}}$ (Eqn. 8) are $SE(3)$-invariant

- $\texttt{MUL}(\mathbf{R}_{\mathcal{AB}}, \mathbf{P}_{\mathcal{B}})$ is $SE(3)$-equivariant with respect to $\mathbf{P}_{\mathcal{B}}$

- $\texttt{PRO}(\mathbf{P}_{\mathcal{A}}, \tilde{\mathbf{P}}_{\mathcal{A}})$ is left- and right- $SE(3)$-equivariant with respect to its inputs $\mathbf{P}_{\mathcal{A}}, \tilde{\mathbf{P}}_{\mathcal{A}}$, respectively, in the sense defined by Equation 24.

If $f_{\theta_{\mathcal{A}}}, f_{\theta_{\mathcal{B}}}$ are $SE(3)$-invariant, then the features $\Phi_{\mathcal{A}}$ and $\Phi_{\mathcal{B}}$ output by these functions do not change with any transformations of the input; we thus refer to $\Phi_{\mathcal{A}}$ and $\Phi_{\mathcal{B}}$ as "invariant features." Thus, $\mathbf{R}_{\mathcal{AB}}$ (Eqn. 12) must also be an $SE(3)$-invariant feature, because it is a function exclusively of invariant features $\Phi_{\mathcal{A}}$ and $\Phi_{\mathcal{B}}$.

Therefore:

$$
\begin{aligned}
f(T_{\mathcal{A}} \circ \mathbf{P}_{\mathcal{A}}, T_{\mathcal{B}} \circ \mathbf{P}_{\mathcal{B}}) &= \texttt{PRO}\left(T_{\mathcal{A}} \circ \mathbf{P}_{\mathcal{A}}, \texttt{MUL}(\mathbf{R}_{\mathcal{AB}}, T_{\mathcal{B}} \circ \mathbf{P}_{\mathcal{B}})\right) \\
&= \texttt{PRO}\left(T_{\mathcal{A}} \circ \mathbf{P}_{\mathcal{A}}, T_{\mathcal{B}} \circ \texttt{MUL}(\mathbf{R}_{\mathcal{AB}}, \mathbf{P}_{\mathcal{B}})\right) \\
&= T_{\mathcal{B}} \circ \texttt{PRO}(\mathbf{P}_{\mathcal{A}}, \texttt{MUL}(\mathbf{R}_{\mathcal{AB}}, \mathbf{P}_{\mathcal{B}})) \circ T_{\mathcal{A}}^{-1} \\
&= T_{\mathcal{B}} \circ f(\mathbf{P}_{\mathcal{A}}, \mathbf{P}_{\mathcal{B}}) \circ T_{\mathcal{A}}^{-1}
\end{aligned}
$$

where the first line follows from the definition of $f$ in Eqn. 23 and because $\mathbf{R}_{\mathcal{AB}}$ is an invariant feature as explained above; the second line follows from the second assumption; the third line follows from the third assumption; and the fourth line is just the definition of $f$ in Eqn. 23. $\square$

As noted in the main text, to make $f_{\theta_{\mathcal{A}}}, f_{\theta_{\mathcal{B}}}$ to be $SE(3)$-invariant, we can either use Vector Neuron layers Deng et al. (2021) or we can train a standard network to produce $SE(3)$-Invariant representations, although the resulting features would not be guaranteed to be invariant. We consider both approaches in our experiments.

The function $\texttt{MUL}$ is discussed in Sec. 5.3 and defined formally in Appendix A.1. As can be seen, this function consists of linear operations and thus is trivially $SE(3)$-equivariant with respect to its second input.

The third assumption follows by the linear algebra properties of Singular Value Decomposition; following the steps of Algorithm 2, if we compute $\texttt{PRO}(T_{\mathcal{A}} \circ \mathbf{P}, T_{\mathcal{B}} \circ \mathbf{Q})$, then $\mathbf{X}, \mathbf{Y}$ will be transformed to $R_A \cdot \mathbf{X}, R_B \cdot \mathbf{Y}$ respectively, where $R_A, R_B$ are the rotation components of $T_{\mathcal{A}}, T_{\mathcal{B}}$, since $\texttt{PRO}$ first subtracts off the mean of $\mathbf{P}$ and $\mathbf{Q}$. Since SVD computes a set of matrices $(\mathbf{U}, \mathbf{\Sigma}, \mathbf{V}^T)$ where $\mathbf{U}$ and $\mathbf{V}$ are rotation matrices, then $\mathbf{U}$ and $\mathbf{V}$ will then be transformed to $R_A \cdot \mathbf{U}$ and $R_B \cdot \mathbf{V}$. We then compute the rotation matrix $R$ that is output by the algorithm as $\mathbf{V}\mathbf{D}\mathbf{U}^T$, so the output is then transformed to $R_B \cdot R \cdot R_A^T$, which is equivalent to $R_B \cdot R \cdot R_A^{-1}$. The translation component is similarly transformed, which leads us to the property that

$$
\texttt{PRO}(T_{\mathcal{A}} \circ \mathbf{P}, T_{\mathcal{B}} \circ \mathbf{Q}) = T_{\mathcal{B}} \circ \texttt{PRO}(\mathbf{P}, \mathbf{Q}) \circ T_{\mathcal{A}}^{-1} \tag{24}
$$

## C  TRAINING DETAILS

We present various implementation details for our method.

### C.1  NETWORK ARCHITECTURES

**Point-Cloud Encoders**: We use standard segmentation-style DGCNN Wang et al. (2019) and VN-DGCNN Deng et al. (2021) (DGCNN with Vector Neuron layers) architectures for our encoders. Pytorch implementations with identical network structure can be found (DGCNN) here and (VN-DGCNN) here.

**Cross-Attention Layer**: We implement a single multi-head attention block with 4 heads, with a similar network structure as implemented in Pytorch here.

**Kernel Network**: Our implementation of the kernel network is a small MLP with 2 hidden layers of size $[300, 100]$, ReLU activation, and BatchNorm.

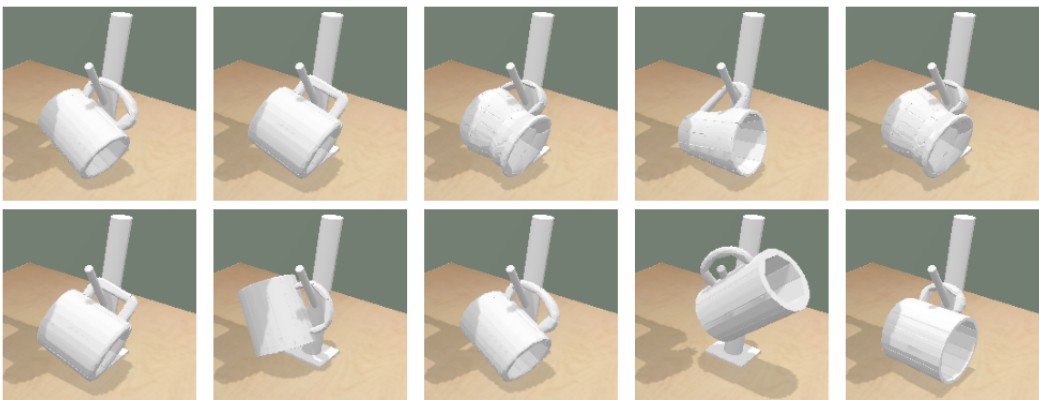

Figure 6: Examples from the dataset of mugs used to train our method. Notice that the mugs vary in geometry, but have similar overall structure.

## C.2 SAMPLING IN MULTILATERATION

In Equation 12, we described $\mathbf{R}_{\mathcal{AB}}$ as an $N_\mathcal{A} \times N_\mathcal{B}$ matrix. In practice, $N_\mathcal{A}$ and $N_\mathcal{B}$ can be quite large ($> 1000$), and each pair must be passed into an MLP to evaluate the kernel. For sufficiently large point clouds, this leads to an explosion in GPU memory, where even a single training example will not fit in GPU memory[4]. To address this issue, we instead sample $K$ points from each point cloud $\mathbf{P}_\mathcal{A}$ and $\mathbf{P}_\mathcal{B}$ to construct a $K \times K$ kernel matrix. In this work, we sample uniformly at random. Importantly, this sampling happens **only** during the construction of the kernel matrix - everything upstream (encoders and cross-attention) are computed on the full-resolution point clouds. In our results, we report a model where $K = 256$, and we show an ablation in Appendix F an example of performance where $K = 100$.

## C.3 PRETRAINING

Following Pan et al. (2023), we pretrain our encoders (both DGCNN and VN-DGCNN) to produce $SE(3)$-invariant encodings using contrastive learning. This effectively learns to map each unique region of a mug, for instance, to a unique representation space (similar to Dense Object Nets Florence et al. (2018) representations). The pretraining involves sampling from $SE(3)$, yielding an $SE(3)$-invariant DGCNN encoder. However, our VN-DGCNN is already $SE(3)$-invariant, so this pretraining step has effect of producing contrastive embeddings which improve performance and convergence speed when training our method using VN-DGCNN encoders. See Appendix F for a comparison of how pretraining affects model performance.

## C.4 SUPERVISION

We use identical supervision as proposed in Pan et al. (2023), combining the Point Displacement Loss, Direct Correspondence Loss, and Correspondence Consistency Loss. See Pan et al. (2023) for details. Examples of training mugs can be found in Figure 6

---

[4]Our implementation may not be perfectly efficient with its GPU utilization, but the $N^2$ relationship is fundamentally a bottleneck for this method.

## D  ADDITIONAL RLBENCH EXPERIMENTS

**Ablation: Sample Efficiency**: To understand how the number of training demonstrations affects the precision of our method, we train variants of our method and TAX-Pose with 1, 5, and 10 training demos from the `stack_wine` task. Results are found in Table 7. We find that our method achieves very high levels of precision with only a single demonstration, and has substantially higher sample efficiency than TAX-Pose.

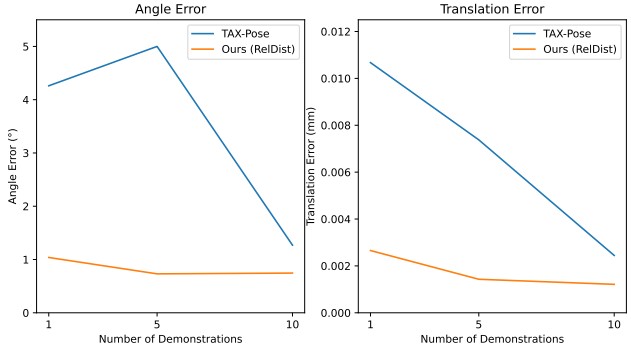

Figure 7: A comparison between number of training demonstrations and prediction precision on the `stack_wine` task.

## E  ADDITIONAL NDF TASK INFORMATION AND RESULTS

### E.1  EVALUATION METRICS

In the original NDF and TAX-Pose papers, the performance of each method is evaluated based on whether the object is successfully grasped by the gripper ("Grasp Success"), successfully placed in a goal location ("Place Success"), and both successfully grasped and successfully placed in the same episode ("Overall Success").

When considering the *precision* of a pose prediction method for a task, however, there is one critical detail of the original evaluation metric which impedes our ability to measure precision: specifically, in the original evaluation protocol, placement success is measured by teleporting an object (without collision) to its pre-

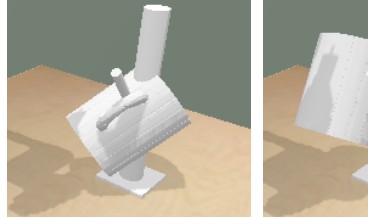
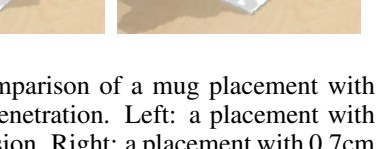

Figure 8: Comparison of a mug placement with and without penetration. Left: a placement with 3.4cm of collision. Right: a placement with 0.7cm of collision.

dicted pose, enabling collisions, stepping a physics simulator for 5 seconds, and detecting whether or not the object has hit the floor at the end of this period. This gives rise to the following degenerate case: if a prediction is made which would result in an intersection of the object with the gripper or environment (for instance, the mug is in direct intersection with the base of the rack), the object may simply stick in place because of the intersection and not fall to the ground (despite being geometrically invalid). See Figure 8 for a visualization of a geometrically invalid and valid placement, both of which are measured as a "success" in the original paper.

To get some sense of how precise these methods actually are, we propose a simple modification to the success metric: after the mug has been teleported to a placement location, we detect whether any penetrations between the mug and the rack have occurred (a trivial computation in our simulated environment), and mark the placement as a failure if the penetration magnitude exceeds a certain threshold (we evaluate on thresholds $1cm, 3cm,$ and unbounded). Like before, we also mark the placement as a failure if the object falls on the floor after being released by the gripper.

| **Mug Hanging** (Upright/Arbitrary) | | | | | | | |
|---|---|---|---|---|---|---|---|
| | **Grasp** | **Place** | | | **Overall** | | |
| Method | | <1cm | <3cm | ∞ | <1cm | <3cm | ∞ |
| DON[5] | 0.91/0.35 | - | - | 0.50/0.45 | - | - | 0.45/0.17 |
| NDF [6] | 0.96/0.78 | - | - | 0.92/0.75 | - | - | 0.88**/0.58** |
| TAX-Pose [7] | **1.00**/0.50 | 0.15/0.32 | 0.70**/0.75** | 0.93/0.85 | 0.15/0.14 | 0.70**/0.40** | 0.93/0.44 |
| Ours | **1.00**/0.43 | **0.35/0.36** | **0.78**/0.66 | **0.99/0.88** | **0.35/0.19** | **0.78**/0.32 | **0.99**/0.41 |
| **Bowl Placement** (Upright/Arbitrary) | | | | | | | |
| | **Grasp** | **Place** | | | **Overall** | | |
| Method | | <1cm | <3cm | ∞ | <1cm | <3cm | ∞ |
| DON | 0.50/0.08 | - | - | 0.35/0.20 | - | - | 0.11/0.00 |
| NDF | 0.91/0.79 | - | - | **1.00/0.97** | - | - | **0.91/0.78** |
| TAX-Pose | **0.87/0.66** | **0.47/0.40** | **0.53/0.56** | 0.57/0.64 | **0.37/0.26** | **0.43/0.37** | 0.47/0.44 |
| Ours | 0.67/0.59 | 0.36/0.19 | 0.49/0.42 | 0.66/0.68 | 0.26/0.08 | 0.35/0.23 | 0.46/0.35 |
| **Bottle Placement** (Upright/Arbitrary) | | | | | | | |
| | **Grasp** | **Place** | | | **Overall** | | |
| Method | | <1cm | <3cm | ∞ | <1cm | <3cm | ∞ |
| DON | 0.79/0.05 | - | - | 0.24/0.02 | - | - | 0.24/0.01 |
| NDF | 0.87/0.78 | - | - | **1.00/0.99** | - | - | **0.87/0.77** |
| TAX-Pose | 0.24/0.16 | **0.89/0.58** | **0.92/0.65** | 0.99/0.72 | **0.22/0.14** | 0.23/0.14 | 0.24/0.14 |
| Ours | **0.65/0.42** | 0.25/0.19 | 0.47/0.42 | 0.52/0.53 | 0.17/0.08 | **0.35/0.22** | 0.37/0.25 |

Table 3: NDF Tasks, Success Rate (↑ is better): The success rate of each method when running the relative placement tasks with the objects starting either upright on the table (left number) or in arbitrary poses above the table (right number). Each column group shows the success rates for the **Grasp** phase and **Place** phase, as well as **Overall** performance (when both grasp and place are successful in a trial). Additionally, for **Place** and **Overall** we report success when maximum penetration is thresholded at the given distance (1cm, 3cm, ∞). In Mug hanging, our method outperforms nearly every baseline.

## E.2 BASELINES

We compare our method to the following baselines:

- Dense Object Nets Florence et al. (2018): Dense Object Nets learns a representation which is viewpoint-invariant, and then attempts to match a novel image with an existing demonstration. The approach to using Dense Object Nets to solve the relative placement task is given in Simeonov et al. (2022).
- Neural Descriptor Fields Simeonov et al. (2022): Learns an SE(3)-equivariant field, which is then optimized at test time to match a set of demonstrations.
- TAX-Pose Pan et al. (2023): This approach shares some similarities with ours, except that this approach learns a residual flow field which is not SE(3)-invariant by construction, instead of our method which estimates pairwise distances.

## E.3 ADDITIONAL EXPERIMENTS

In this section, we present additional simulated results on two tasks presented in the NDF paper Simeonov et al. (2022): Bottle Placement and Bowl Placement. Training datasets for these tasks are

generated in an identical fashion to the NDF Mug-Hanging task described in Section 6, as are the evaluation procedures and metrics.

**NDF Bottle Placement Task**: In this task, the robot must grasp a bottle, and place it on a shelf in its workspace. The bottle is frequently symmetric. Following Pan et al. (2023), we add an additional color channel to the object to break this symmetry.

**NDF Bowl Placement Task**: Similar to bottle, in this task, the robot must grasp a bowl, and place it on a shelf in its workspace. The bowl is frequently symmetric. Following Pan et al. (2023), we add an additional color channel to the object to break this symmetry.

### E.4 ANALYSIS

We find that our method performs comparably to the closest baseline, TAX-Pose, when trained using the exact code and training procedure provided by Pan et al. (2023). However, performance for both our retrained version of TAX-Pose and our own method are substantially worse (by roughly the same amount) than the original reported TAX-Pose results - this is primarily due to the fact that no implementation of symmetry-breaking technique described in Pan et al. (2023) was provided in the initial code release, and we were not able to reproduce reported results with our own implementation. Pan et al. (2023) report this technique as being very important for good performance. We believe that additional details are needed to reproduce the strong performance reported in Pan et al. (2023), and will update this manuscript once we obtain the implementation from the authors.

## F ABLATIONS

We further perform the following ablations to understand the effect of changing different parts of our method. Ablation results can be found in Tables 4, 5, and 6.

| Mug Hanging ($\mathbf{z/SE(3)}$) | | | | | | | |
|---|---|---|---|---|---|---|---|
| | **Grasp** | **Place** | | | **Overall** | | |
| Method | | <1cm | <3cm | $\infty$ | <1cm | <3cm | $\infty$ |
| TAX-Pose, no SE(3) | 0.29 | 0.01 | 0.35 | 0.37 | 0.00 | 0.05 | 0.06 |
| Ours, no SE(3) | **0.35** | **0.24** | **0.77** | **0.87** | **0.05** | **0.27** | **0.31** |

Table 4: Ablation: Removing SE(3) Data Augmentation - Arbitrary Mug Hanging, Success Rate ($\uparrow$ is better): Compared to the standard training proposed by TAX-Pose (Pan et al. (2023)), we remove all SE(3) data augmentations from the training pipeline. We train only on demonstration poses $\mathbf{z}$ directly (which are mostly vertical). TAX-Pose performance degrades significantly, while our method's performance remains strong.

**Impact of SE(3) Data Augmentations**: In the original training procedure for TAX-Pose (Pan et al. (2023)), the authors introduce substantial data augmentation in the form of random SE(3) transforms to inputs in order to achieve neural networks which are approximately SE(3)-equivariant. To demonstrate that our method can achieve high success rates *without* these complex data augmentations, we train a version of both our provable-equivariant model, as well as the standard TAX-Pose model. We report our findings in Table 4. We find that while TAX-Pose fails entirely to generalize to mugs in novel configurations, because our method is provably SE(3)-Equivariant, it demonstrates strong performance across SE(3) even when trained on mugs in a narrow range of starting poses.

**Pretraining**: We evaluate the effect of pretraining on performance for both tasks. The optimal setting of $f_{\theta_{\mathcal{A}}}, f_{\theta_{\mathcal{B}}}$ are $SE(3)$-Invariant, which the pretraining process approximately initializes for DGCNN (our VN-DGCNN is guaranteed to be invariant). In Pan et al. (2023), the authors show that pretraining the standard DGCNN encoder improves test-time performance. However, we want to see whether or not this pretraining procedure improves performance for the VN-DGCNN version (which is guaranteed to be invaraint). Intuitively, this pretraining would simply initialize the representation for all points on each object to be far from one another. Interestingly, in addition to helping training converge faster, pretraining substantially improves performance.

**Sampling**: We evaluate whether the value of $K$ in the sampling procedure for computing the kernel matrix $\mathbf{R}_{\mathcal{AB}}$ detailed in Appendix C affects performance (see "VN-DGCNN ($K = 100$)" in Tables 5 and 6). We find that increasing the number of sampled points improves the performance, which intuitively suggests that the multilateration optimization will be able to reduce the variance of predictions (and thus increase prediction precision) with more samples.

**Vector Neurons**: We evaluate how training a standard DGCNN network to predict $SE(3)$-invariant features for each object encoder compares to using the fully $SE(3)$-invariant VN-DGCNN to predict those same features. We find that the standard DGCNN network is superior on some precision metrics, while VN-DGCNN has superior overall performance. Because we are using networks of a similar size, this suggests that there may a representational penalty for limiting a network to be fully-equivariant. Additionally, many of the mugs are perfectly or nearly symmetrical - this means that the VN-DGCNN network may not have sufficient representational flexibility to differentiate between two instances, i.e. to "tiebreak" depending on the object's initial orientation.

| Method | Grasp | Place | | | | | Overall | | | | |
|---|---|---|---|---|---|---|---|---|---|---|---|
| | | =0cm | <1cm | <2cm | <3cm | $\infty$ | =0cm | <1cm | <2cm | <3cm | $\infty$ |
| DON | 0.91 | - | - | - | - | 0.50 | - | - | - | - | 0.45 |
| NDF | 0.96 | - | - | - | - | 0.92 | - | - | - | - | 0.88 |
| TAX-Pose (release) | **1.00** | 0.01 | 0.07 | 0.26 | 0.61 | 0.95 | 0.01 | 0.07 | 0.26 | 0.61 | 0.95 |
| TAX-Pose (retrained) | **1.00** | 0.04 | 0.24 | 0.44 | 0.67 | 0.96 | 0.04 | 0.24 | 0.44 | 0.66 | 0.96 |
| VN-DGCNN ($K$=100) | 1.00 | 0.02 | 0.16 | 0.36 | 0.64 | 0.97 | 0.02 | 0.16 | 0.36 | 0.64 | 0.97 |
| VN-DGCNN (no-pretrain) | 0.99 | 0.03 | 0.21 | 0.49 | 0.79 | 0.93 | 0.03 | 0.20 | 0.48 | 0.78 | 0.92 |
| DGCNN | 0.98 | **0.05** | **0.31** | **0.64** | **0.83** | 0.97 | **0.05** | **0.30** | **0.63** | **0.82** | 0.96 |
| VN-DGCNN (Ours) | **1.00** | 0.03 | 0.28 | 0.57 | 0.80 | **0.98** | 0.03 | 0.28 | 0.57 | 0.80 | **0.98** |

Table 5: Ablations - Upright Mug Hanging, Success Rate (↑ is better): The success rate of each method when running the mug-hanging task with the mugs starting upright on the table. Each column group shows the success rates for the **Grasp** phase and **Place** phase, as well as **Overall** performance (when both grasp and place are successful in a trial). Additionally, for **Place** and **Overall** we report success when maximum penetration is thresholded at the given distance. Across nearly all metrics, our method outperforms the baselines.

| Method | Grasp | Place | | | | | Overall | | | | |
|---|---|---|---|---|---|---|---|---|---|---|---|
| | | =0cm | <1cm | <2cm | <3cm | $\infty$ | =0cm | <1cm | <2cm | <3cm | $\infty$ |
| DON | 0.35 | - | - | - | - | 0.45 | - | - | - | - | 0.17 |
| NDF | **0.78** | - | - | - | - | 0.75 | - | - | - | - | 0.58 |
| TAX-Pose (release) | 0.74 | **0.04** | 0.18 | 0.48 | **0.74** | 0.89 | **0.03** | 0.14 | 0.35 | **0.55** | **0.65** |
| TAX-Pose (retrained) | 0.68 | 0.03 | 0.20 | 0.42 | 0.61 | 0.86 | 0.01 | 0.14 | 0.30 | 0.42 | 0.58 |
| VN-DGCNN ($K$=100) | 0.61 | 0.04 | 0.22 | 0.39 | 0.63 | 0.91 | 0.02 | 0.12 | 0.22 | 0.39 | 0.56 |
| VN-DGCNN (no-pretrain) | 0.44 | **0.05** | 0.28 | 0.52 | 0.71 | 0.82 | 0.02 | 0.13 | 0.24 | 0.32 | 0.37 |
| DGCNN | 0.68 | 0.04 | **0.30** | 0.51 | 0.70 | 0.89 | **0.03** | **0.20** | 0.35 | 0.48 | 0.61 |
| VN-DGCNN (Ours) | 0.67 | 0.04 | 0.29 | **0.55** | 0.72 | **0.92** | **0.03** | 0.18 | **0.36** | 0.47 | 0.62 |

Table 6: Ablations - Arbitrary Mug Hanging, Success Rate (↑ is better): The success rate of each method when running the mug-hanging task with the mugs starting in arbitrary orientations above the table.

## G   REAL-WORLD EXPERIMENTS

**Experiment Description**: To demonstrate the ability of our method to generalize to real-world robotics scenarios, we design an experiment that closely follows the real-world experiments proposed in Pan et al. (2023). Specifically, we choose the real-world mug-hanging experiment from this work. The authors from Pan et al. (2023) have graciously provided us with a dataset real-world mug-hanging demonstrations, collected in the setup displayed in Figure 9. This dataset consists of 20 different demonstrations of various mugs placed on a single

| Method | Trans. Error (m) | Rot. Error (°) |
|--------|------------------|----------------|
| **Ours** | 0.1 | 35° |

Table 7: Average prediction error on **unseen**, real-world mugs. The metrics are computed comparing the predicted position of the mug to the actual position of the mug in the demonstration.

rack, where the rack is placed in different places in the workspace of a robot. The demonstrations consist of initial and final observations of the mug and rack. We split this dataset into 10 training mugs and 10 test mugs. We then train our method in the exact same way as we did our simulated mugs, and evaluate on the test mugs.

**Analysis of Results**: Our evaluation is both quantitative and qualitative. First, our quantitative evaluation procedure is to randomly perturb a real test demo capture, and attempt to predict its a success configuration. We then compute a translational and rotational error, to meausre how far the mug was placed relative to the test demonstrations. See Table 7. Because the mug can rotate freely around the peg, the relatively high rotation error is not indicative of poor predictions - in fact, these metrics are frequently within the tolerance for successful mug hanging given scene geometry.

To further qualitatively support this claim, consider real predictions we provide in Figure 11. We see that predictions are consistent with successful mug-hanging, with the handle placed over the peg of the rack. Furthermore, the format of these predictions (a simple rigid transform) is identical to those in Pan et al. (2023), which led in their work to successful robotic placement in a real system. While we do not execute a motion policy based on these predictions on a real-world robot arm, we believe that this qualitative and quantitative evidence provides strong support for our claim that our model could be successfully used in a real-world robot system similar to that proposed by Pan et al. (2023).

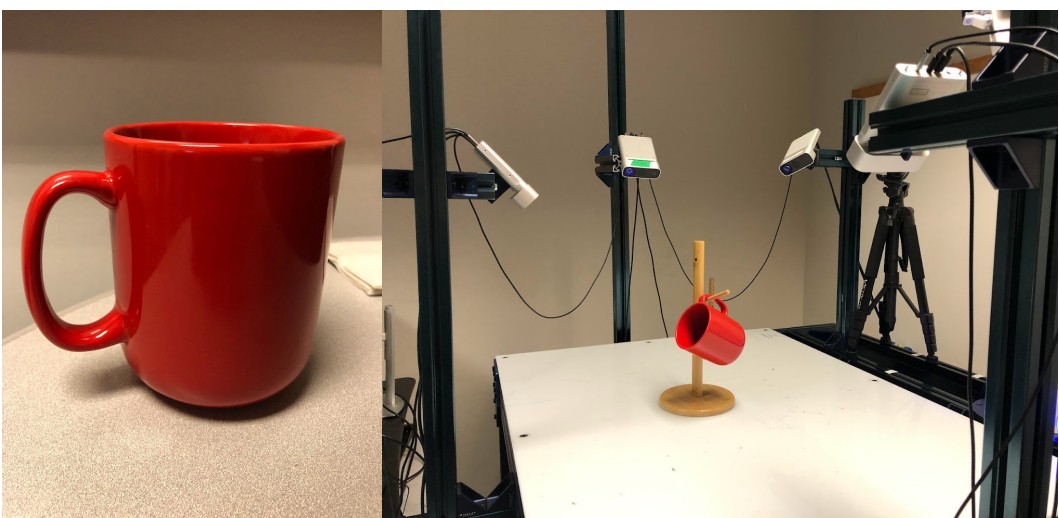

(a) A real training mug                                 (b) A demonstration of mug-hanging

Figure 9: An example of how demonstrations were collected in a real-world robot workspace. Images provided by the authors of Pan et al. (2023).

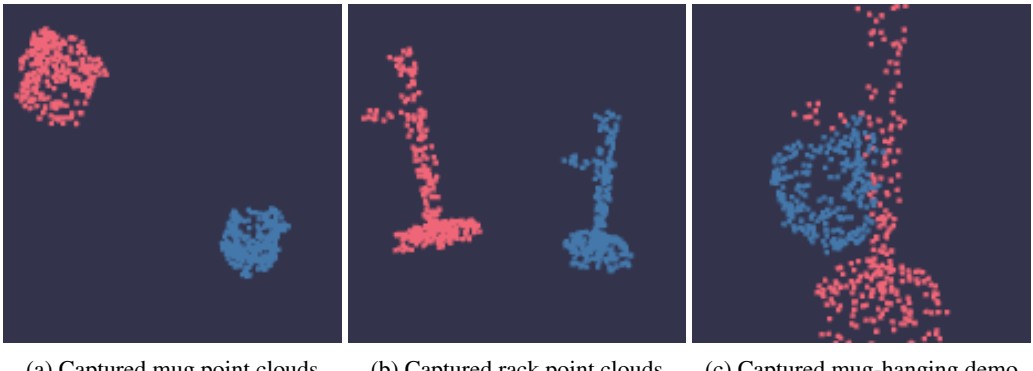

(a) Captured mug point clouds  (b) Captured rack point clouds  (c) Captured mug-hanging demo

Figure 10: Examples of training data collected with the setup in Figure 9. The mugs are segmented from the rack by the procedure outlined in Pan et al. (2023), and then used as training data for our method.

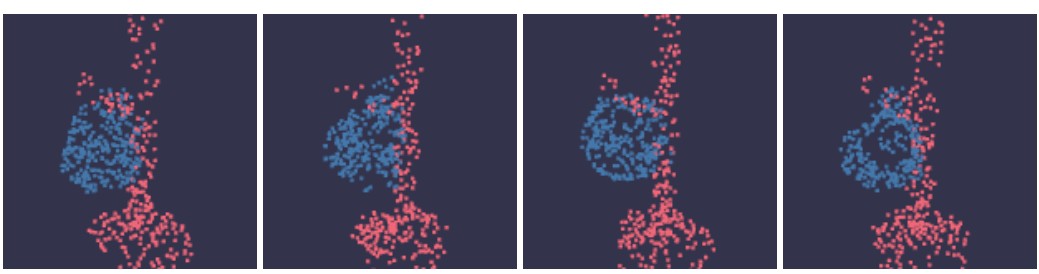

Figure 11: Images of distinct, unseen real-world mugs in their predicted goal configurations. Our method makes high-quality predictions on **novel mug instances** in **unseen configurations** from real-world sensor data. Because we make outputs in the same format as in Pan et al. (2023), we believe that these predictions are strong evidence that our method could successfully be used to solve relative placement tasks on a real-world robot system similar to that proposed in Pan et al. (2023).

