# OpenReview forum: "Deep SE(3)-Equivariant Geometric Reasoning for Precise Placement Tasks"
_ICLR.cc/2024/Conference — ICLR 2024 poster_

### Official Review · Reviewer_Te9m · 2023-11-01

**Soundness:** 2 fair
**Presentation:** 3 good
**Contribution:** 2 fair
**Rating:** 6
**Confidence:** 4

**Summary:**

This paper proposes a novel model for implementing the SE(3)-equivariance in robotic pick and place problems. Given a task involving placing an object relative to another object, the method first uses an equivariant network to calculate the desired distance between points on the two objects. The core innovation of the method is that given the desired distances from a point on object A to a set of points on object B, one can use multilateration to locate the desired location of this point relative to object B. Repeating this process for multiple points on object A, the relative pose between A and B could then be confirmed. The proposed method is evaluated against a number of baselines, where it demonstrates commendable performance, especially when the task requires high precision.

**Strengths:**

1. The idea of using multilateration to calculate the relative pose is a compelling aspect of this paper. It transforms the equivariant problem of calculating the desired relative pose into an invariant problem of calculating the desired relative distance, which could be useful to reduce the complexity of a model.
2. The paper is well-written with intuitive examples to illustrate the idea.

**Weaknesses:**

My main concern with the paper is that the experimental evaluation is not strong enough. In the main paper, the experiments are mainly conducted in the Mug Hanging domain. In the two other domains in Table 3 in the appendix, the proposed method’s performance is worse than the baselines. Though the authors discuss that the underperformance compared with TAX-Pose could be due to the lack of the implementation of the symmetry-breaking technique, the proposed method also lags behind NDF. Additionally, the performance of DON and NDF under 1cm and 3cm distance thresholds is not reported in either Table 1 or Table 3.

**Questions:**

1. As is mentioned in Weakness, my main comment is about the experiment section. The paper would be much stronger if there is an environment that actually requires high place precision (e.g., gear/kit assembly) rather than merely adjusting the precision requirement in the mug-hang task. Moreover, incorporating the performance of NDF under the 1cm and 3cm precision requirements in Tables 1 and 3 would likely strengthen the paper as well.
2. Does the proposed method have a faster runtime compared with the baselines? Does the proposed method have a higher sample efficiency? Since the idea of using multilateration reduces the equivariant problem into an invariant problem, I am curious about if there are benefits from this angle.

---

> ### Author Response · Authors · 2023-11-11
> **Initial Response to Review**
>
> We thank the reviewer for their feedback, and appreciate the reviewer’s recognition of our novel SE(3)-invariant representation and clear presentation.
>
> Addressing the reveiwer’s questions:
>
> > As is mentioned in Weakness, my main comment is about the experiment section. The paper would be much stronger if there is an environment that actually requires high place precision (e.g., gear/kit assembly) rather than merely adjusting the precision requirement in the mug-hang task.
>
> We can certainly attempt to prepare an additional set of high-precision tasks to benchmark the performance of our method against baselines. Specifically, we are considering the following additional experiments:
>
> * 5 relative placement tasks from the RLBench suite of tasks. https://github.com/stepjam/RLBench
>     * Insert Peg (precise placement of disc on peg)
>     * Insert USB (precise placement of USB in computer)
>     * Toilet Roll on Stand
>     * Phone on Base
>     * Wine Bottle on Rack
>
> There are no specific gear/kit assembly tasks in RLBench, but “Insert Peg” is quite close in spirit (and has existing demonstrations we can leverage).
>
> Additionally, we are attempting to rectify the issues we faced with the TAX-Pose symmetry-breaking techniques that suppressed performance. Hopefully that will allow us to present complete results on Bottle and Bowl placement.
>
> **Question for the reviewer: Would including results on these additional tasks satisfy the reviewer’s desire for more high-precision tasks? If not, are there additional specific existing benchmarks that the reviewer would like to see?**
>
> > Moreover, incorporating the performance of NDF under the 1cm and 3cm precision requirements in Tables 1 and 3 would likely strengthen the paper as well.
>
> We agree that this would be a good and interesting evaluation. Time permitting, we will attempt to extract these values by the end of the evaluation period.
>
> > Does the proposed method have a faster runtime compared with the baselines?
>
> Compared to TAX-Pose, we require slightly more compute (due to the additional reasoning layers), but the sample complexity is only a constant factor difference. Compared to NDF, which uses an iterative optimization, we execute significantly faster since we only require a single forward pass of the network to regress.
>
> > Does the proposed method have a higher sample efficiency?
>
> Because the network is SE(3) equivariant, we can learn per-object relationships from very few samples (potentially even from a single example). As for class-level relationships (i.e. generalizing to all mugs instead of a single mug), we will attempt an experiment which is similar to that presented in Table 2 of TAX-Pose, i.e. seeing how success rate changes with number in-class demonstrations.
>
> We will provide another update with experiments as they are ready during the rebuttal period.

---

> > ### Comment · Reviewer_Te9m · 2023-11-14
> >
> > Thank you for the quick response to my comments. I think all the proposed experiments can strengthen the paper. Including the five (especially the first two) proposed environments from RLBench would address my concern about high-precision task.

---

> ### Author Response · Authors · 2023-11-22
> **New experiments: 5 high-precision tasks on RLBench, our method outperforms baselines**
>
> We have completed new experiments on 5 high-precision tasks from the RLBench suite - see our Official Comment posted at the top. To summarize, our method achieves very high levels of precision (<1.25 degrees of rotational error, <2mm of translational error on average) on every task, which is within the tolerance margin for successful predictions for each task. The baseline we compare against, TAX-Pose, does not achieve this level of precision at convergence.
>
> We switched out the “Insert USB” task because we found that the demo-generating code for this task was broken in the RLBench code release, yielding no demos on which to train. Instead, we have chosen a different high-precision task, “Place Hanger on Rack”, where there is only ~1cm of clearance.
>
> **Given that our method achieves a high degree of precision on these 5 new high-precision tasks, would you consider increasing your score?**
>
> We are also in the process of conducting additional experiments + writing to address your other concerns, and will post again tomorrow.

---

> > ### Comment · Reviewer_Te9m · 2023-11-22
> >
> > Thank you for the new experimental results, they seem pretty strong. I will consider increasing my score upon reading the revision and your final response. Looking forward to it!

---

> > > ### Author Response · Authors · 2023-11-23
> > > **Final update (New experiments, new manuscript)**
> > >
> > > We thank the reviewer for acknowledging the strength of our additional experiments. We wanted to summarize the new results during the rebuttal period, including two new sets of requested experiments and updates to our manuscript.
> > >
> > > # New Experiments
> > >
> > > During the rebuttal period, we performed the following experiments:
> > >
> > > * We evaluated our method on a new set of 5 precise placement tasks from the RLBench task suite. We found that our method achieves **substantially higher precision** than the closest baseline, TAX-Pose. We believe that this is strong evidence of our method’s suitability for precise placement tasks.
> > >
> > > * We have performed an additional **sample efficiency** ablation. See our second post above, entitled “New experiment: Strong sample efficiency for learning precise placement”. We find that our method is able to learn a highly-precise relationship from only a single demonstration, and uniformly achieves higher precision-per-sample than TAX-Pose.
> > >
> > > # Manuscript Revisions
> > >
> > > We have made the following changes to the manuscript:
> > >
> > > * Added a new experiment subsection, which describes the RLBench suite of tasks and presents our main results and includes some visualizations of the task.
> > > * Added the new sample efficiency ablation
> > > * Created a new original figure illustrating relative placement for the Problem Statement section.
> > >
> > > ***Additional Comments***
> > >
> > > We did not have time to resolve the symmetry issues that affected the bottle/bowl tasks in both our TAX-Pose and RelDist evaluations. We did discover that our implementation of symmetry-breaking has subtle errors which results in large ambiguity in the regression target - so the performance reported in the Appendix is not representative of TAX-Pose/our performance on these tasks. We are confident that we will be able to rectify the symmetry-breaking bug we currently are experiencing and report accurate results on those two tasks prior to final publication. However, we believe that the performance of our method on the remaining 6 tasks reported (the NDF Mug task as well as the 5 new RLBench tasks) are strong enough on their own.
> > >
> > > Unfortunately, we did not have time in the rebuttal period to evaluate NDF’s precision performance (a metric we introduced) on the NDF benchmark tasks, or conduct any real-world robot experiments.
> > >
> > > **Overall, we hope we have addressed the majority of the reviewers’ concerns about our experiments section. We believe that our revised manuscript is substantially stronger with the addition of RLBench experiments and analysis.**

---

### Official Review · Reviewer_5kHb · 2023-11-01

**Soundness:** 2 fair
**Presentation:** 2 fair
**Contribution:** 2 fair
**Rating:** 6
**Confidence:** 4

**Summary:**

This paper propose a method for precise relative pose prediction which is provably SE(3)-equivariant, can be learned from only a few demonstrations, and can generalize across variations in a class of objects. The core technical contribution is to transform the optimization of two SE(3) fields into a differentiable multilateration problem.

**Strengths:**

- This paper proposes a method that tackles SE(3)-equivariant learning by estimating the  corresponding point pairs with differentiable multilateration. The method provides the community with some fresh new ideas.
- This paper is in general written in a clear way, and is easy to comprehend.

**Weaknesses:**

- Presentation
    - The problem statement section as well as some figures are directly borrowed from TAX-Pose. I think it severely damages the presentation of this paper.
- Performance
    - The result in Table 1 is margianl improvement compared with PAX-Pose, though the proposed method does exhibit some advantages in higher-precision settings.
- Real-world experiments
    - This method is evaluated with offline real-world trajectories collected by TAX-Pose. I won't accept this as real-world experiments.
    - In figure, the rotational error is 35 degrees. While the authors have made some explanations, I won't regard it as a satisfactory result.

**Questions:**

- Will there be genuine real-world experiments conducted in the future?

---

> ### Author Response · Authors · 2023-11-11
> **Initial Response to Review**
>
> We thank the reviewer for their feedback, and appreciate the reviewer’s recognition of the novelty of using multilateration for reasoning and the clarity of presentation.
>
> Responding to individual points raised by the reviewer:
>
> > The problem statement section as well as some figures are directly borrowed from TAX-Pose. I think it severely damages the presentation of this paper.
>
> We did reproduce some elements of the problem statement (including one figure) with permission from TAX-Pose authors. We did this to emphasize that the problem setting is nearly identical to theirs; however, we will reword certain sections to differentiate a bit. And we will create an original figure by the end of the rebuttal period to incorporate into the revised manuscript.
>
> > The result in Table 1 is margianl improvement compared with PAX-Pose, though the proposed method does exhibit some advantages in higher-precision settings.
>
> We agree that the “overall” improvements appear somewhat small compared to TAX-Pose; however, the original evaluation procedure proposed by NDF and utilized by TAX-Pose is not particularly robust to geometrically-infeasible predictions, which means that predictions that look poor may still be counted as a success because of peculiarities of simulation modeling. That is why we introduced the “precision” metric, to quantitatively demonstrate that our predictions result in higher-quality predictions than TAX-Pose that respect geometric constraints (and are thereby more precise) more frequently.
>
> > This method is evaluated with offline real-world trajectories collected by TAX-Pose. I won't accept this as real-world experiments.
>
> This is a reasonable criticism. We included this experiment to demonstrate that our method could be trained on real-world data and produce reasonable predictions - however we stopped short of executing these predictions on a real robot system for specific implementation issues mentioned below.
>
> > In figure, the rotational error is 35 degrees. While the authors have made some explanations, I won't regard it as a satisfactory result.
>
> We agree that this rotational error appears unsatisfactory. The metric we chose to present, rotational error w.r.t. A held-out set of demonstrations, does not inherently capture the multimodality of the demonstration set. There are many viable predictions that result in task success, which may differ dramatically in rotation from those seen in demonstrations. For instance, hanging the mug to the left side and to the right side are both valid (and appear in the demonstration set). A better evaluation would be task success rate, where we measure how frequently the predictions lead to a good placement on a real robot system.
>
> > Will there be genuine real-world experiments conducted in the future?
>
> We do have a robot system in our lab with 4 cameras and a robot arm, and before submission did attempt to build a system which would execute the predictions made by our method. We were able to get predictions, however unfortunately we struggled to calibrate the robot arm’s motion to the cameras to a degree of precision which is necessary to execute the predictions open-loop with motion planning. We were able to get 2-3cm end-effector precision in our system, which is not sufficient to place mugs on racks effectively. This was independent of our RelDist prediction quality; it was purely an engineering issue.
>
> Because of the complexity of increasing end-effector precision in a calibrated 4-camera setup, we don’t believe we will be able to produce robot experiments by the end of the rebuttal period (in 12 days). However it is definitely on our roadmap to build a robust real-world evaluation setup eventually which has the hand-eye calibration precision required.
>
> **Question for the reviewer: Are there any real-world experiments you might accept that don’t require executing the RelDist predictions on a real robot?**

---

> > ### Comment · Reviewer_5kHb · 2023-11-13
> > **On Real-World Experiments**
> >
> > I would accept the authors' explanation that high-precision hand-eye calibration is a non-trivial process. But from my perspective, real-world experiments have to be executed on the physical robot. There is no compromise.
> >
> > Additionally, I agree with Reviewer Te9m that *the paper would be much stronger if there is an environment that actually requires high place precision (e.g., gear/kit assembly)*. So if the authors believe they are unable to produce real-world experiments due to calibration issues, I would suggest they make efforts to conduct the experiments required by Reviewer Te9m in the rebuttal period.

---

> ### Author Response · Authors · 2023-11-22
> **New experiments: 5 high-precision tasks on RLBench, our method outperforms baselines**
>
> We have completed new experiments on 5 high-precision tasks from the RLBench suite - see our Official Comment posted at the top. To summarize, our method achieves very high levels of precision (<1.25 degrees of rotational error, <2mm of translational error on average) on every task, which is within the tolerance margin for successful predictions for each task. The baseline we compare against, TAX-Pose, does not achieve this level of precision at convergence.
>
> **Given that our method achieves a high degree of precision on these 5 new high-precision tasks, would you consider increasing your score?**
>
> We are also in the process of conducting additional experiments + writing to address your other concerns, and will post again tomorrow.

---

> > ### Comment · Reviewer_5kHb · 2023-11-23
> > **Reply to New Experiments; Raising Score**
> >
> > Dear authors,
> >
> > I decide to raise my score from 5 to 6, given the newly added experiments during the rebuttal period, and your promise to improve the presentation in the revised version.
> >
> > The reason why I cannot give a higher score is that my primary concern about the real-world experiments has not been resolved.

---

> > > ### Author Response · Authors · 2023-11-23
> > > **Final update (New experiments, new manuscript)**
> > >
> > > We thank the reviewer for acknowledging the strength of our additional experiments, and for raising the score of their review. We wanted to provide a final update, including a new set of requested experiments and updates to our manuscript.
> > >
> > > # New Experiments
> > >
> > > During the rebuttal period, we performed the following experiments:
> > >
> > > * We evaluated our method on a new set of 5 precise placement tasks from the RLBench task suite. We found that our method achieves **substantially higher precision** than the closest baseline, TAX-Pose. We believe that this is strong evidence of our method’s suitability for precise placement tasks.
> > >
> > > * We have performed an additional **sample efficiency** ablation. See our second post above, entitled “New experiment: Strong sample efficiency for learning precise placement”. We find that our method is able to learn a highly-precise relationship from only a single demonstration, and uniformly achieves higher precision-per-sample than TAX-Pose.
> > >
> > > # Manuscript Revisions
> > >
> > > We have made the following changes to the manuscript:
> > >
> > > * Added a new experiment subsection, which describes the RLBench suite of tasks and presents our main results and includes some visualizations of the task.
> > > * Added the new sample efficiency ablation
> > > * Created a new original figure illustrating relative placement for the Problem Statement section.
> > >
> > > ***Additional Comments***
> > >
> > > Unfortunately, we did not have time in the rebuttal period to conduct any real-world robot experiments. This is a top priority for us for future iterations of this manuscript. Because our output format is identical to TAX-Pose (which demonstrated successful mug-hanging on a robot system), and because we demonstrate high-precision predictions on the new RLBench dataset, we believe that we will be able to get this working on a real robot once we are able to overcome our calibration issues. Of course we respect that this may be insufficient to alleviate your concerns at the moment.
> > >
> > > Additionally, we made some minor tweaks to the Problem Statement section, including make a new figure to replace the one we reproduced with permission from TAX-Pose. The wording is quite different from TAX-Pose's original problem statement, although the structure is similar. Additionally, we have decided to retain the reproductions of equations 5 and 6 from TAX-Pose, as they are essential definitions for understanding the task.
> > >
> > > **Overall, we hope we have addressed the majority of the reviewers’ concerns about our simulated experiments section. We believe that our revised manuscript is substantially stronger with the addition of RLBench experiments and analysis.**

---

### Official Review · Reviewer_5GvH · 2023-11-04

**Soundness:** 3 good
**Presentation:** 3 good
**Contribution:** 3 good
**Rating:** 6
**Confidence:** 3

**Summary:**

The paper presents a system that is provably SE(3)-Equivariant for predicting task-specific object poses for relative placement tasks. It introduces a new cross-object representation called RelDist, an SE(3)-Invariant geometric reasoning framework, and employs multilateration and Singular Value Decomposition (SVD) to extract relative pose predictions. The study validates the representation’s performance through simulated high-precision tasks and real-world manipulation demonstrations, emphasizing its applicability to point cloud data.

**Strengths:**

The paper's originality lies in its novel representation for cross-object relationships and the formulation of a problem-solving approach that is SE(3)-Equivariant. The quality of the work is high, evidenced by the clear methodology and promising experimental results. The clarity of the presentation is commendable, with complex concepts and processes being explained with precision. The experiment results are supportive to the precision requirement of the tasks and explained by the algorithm design. Lastly, the significance of the work is underlined by its practical applications in robotics and potential to influence future research in the area.

**Weaknesses:**

Although the experimental results are positive and supportive to the claims on e.g., precision and the algorithm design, more tasks or scenarios could be evaluated. They can still be precise pick and place but should at least be different sets of objects, such as peg-in-hole.  The methodology looks pretty promising (equivariant + differentiable optimization process) and generic, as a submission to a ML conference, one would expect to see a more diverse evaluation of the approach.

Other weaknesses identified pertain to limitations in handling symmetric objects or multimodal placement tasks, as acknowledged by the authors. This could restrict the system's application in scenarios where multiple correct poses exist. Additionally, the requirement for segmented task-relevant objects can be a significant limitation in unstructured environments. The paper could be improved by exploring these aspects further, possibly by integrating generative models or unsupervised segmentation methods.

However, overall, I would still think the paper is slightly above the threshold, while a more complete evaluation could well strengthen the paper.

**Questions:**

- Could experimental results be further augmented with a more diverse set of tasks or scenarios? What could be the choices?
- How does the system handle noisy data or incomplete point clouds, common in real-world scenarios?
- Could the authors elaborate on potential strategies for overcoming the limitations related to symmetric objects or multimodal placements?

---

> ### Author Response · Authors · 2023-11-11
> **Initial Response to Review**
>
> We thank the reviewer for their feedback, and for recognizing the novelty of our formulation, RelDist, in solving relative placement tasks in a provably SE(3)-Equivariant way. We also appreciate that the reviewer felt that the work was clear and of high-quality.
>
> To address the questions specifically:
>
> > Could experimental results be further augmented with a more diverse set of tasks or scenarios? What could be the choices?
>
> We can certainly attempt to prepare an additional set of tasks to benchmark the performance of our method against baselines. Specifically, we are considering the following additional experiments:
>
> * 5 relative placement tasks from the RLBench suite of tasks. https://github.com/stepjam/RLBench
>     * Insert Peg (precise placement of disc on peg)
>     * Insert USB (precise placement of USB in computer)
>     * Toilet Roll on Stand
>     * Phone on Base
>     * Wine Bottle on Rack
>
> In each of these tasks, we would only evaluate the placement of an object relative to another object (and therefore eliminate the need to do robot control for grasping), which should demonstrate applicability across a wider range of tasks.
>
> Additionally, we are attempting to rectify the issues we faced with the TAX-Pose symmetry-breaking techniques that suppressed performance. Hopefully that will allow us to present accurate results on Bottle and Bowl placement.
>
> **Question for the reviewer: Would including these additional experiments strengthen the paper to your satisfaction?**
>
> If so, we will update the rebuttal later in the rebuttal period with results.
>
> If not, what additional simulated experiments would you like to see?
>
> > How does the system handle noisy data or incomplete point clouds, common in real-world scenarios?
>
> Our method can be trained on partial/occluded point clouds - for instance, in our Real World experiment (Section 6.2), we do use four cameras, but they do not see the entire mug. There is no fundamental reason why the method shouldn’t work under noise/occlusion - the RelDist representation is still valid under those circumstances. However, the training procedure would need to incorporate that distribution (we already have a synthetic occlusion data augmentation, and could feasibly incorporate a viewpoint dropout or noise augmentation).
>
> > Could the authors elaborate on potential strategies for overcoming the limitations related to symmetric objects or multimodal placements?
>
> There are two potential approaches we have considered. The first is empirical symmetry-breaking, similar to what we use in the Bottle/Bowl experiments. You could “paint” each object with a feature field which breaks the symmetry - this painting could be learned generatively to flexibly represent the set - and could be consistent across the class. In this way, the learned kernel function would be able to break the symmetry unambiguously. Precisely formulating this learning process is on our roadmap for future work.
>
> The second is analytical, similar to https://openreview.net/forum?id=psyvs5wdAV which enumerates the symmetry group for each object category. With this privileged information, you could assign the supervision target to be a specific canonical element of that symmetry group (i.e. the RelDist for a particular rotation of a symmetric cube), even if the demonstration is a different element of that group, and transform the predicted canonical representation into the group element seen at training time (to make the losses consistent). However, this requires a lot of assumptions, including the existence of a consistent symmetry group across a class (may not be true, i.e. some bottles are symmetric, some are not).

---

> ### Author Response · Authors · 2023-11-22
> **New experiments: 5 high-precision tasks on RLBench, our method outperforms baselines**
>
> We have completed new experiments on 5 high-precision tasks from the RLBench suite - see our Official Comment posted at the top. To summarize, our method achieves very high levels of precision (<1.25 degrees of rotational error, <2mm of translational error on average) on every task, which is within the tolerance margin for successful predictions for each task. The baseline we compare against, TAX-Pose, does not achieve this level of precision at convergence.
>
> Given that our method achieves a high degree of precision on these 5 new high-precision tasks (and provides a more complete evaluation of the method), would you consider increasing your score?

---

> ### Author Response · Authors · 2023-11-23
> **Final update (New experiments, new manuscript)**
>
> We thank the reviewer again for their feedback. We wanted to provide a final update, including a new set of requested experiments and updates to our manuscript.
>
> # New Experiments
>
> During the rebuttal period, we performed the following experiments:
>
> * We evaluated our method on a new set of 5 precise placement tasks from the RLBench task suite. We found that our method achieves **substantially higher precision** than the closest baseline, TAX-Pose. We believe that this is strong evidence of our method’s suitability for precise placement tasks.
>
> * We have performed an additional **sample efficiency** ablation. See our second post above, entitled “New experiment: Strong sample efficiency for learning precise placement”. We find that our method is able to learn a highly-precise relationship from only a single demonstration, and uniformly achieves higher precision-per-sample than TAX-Pose.
>
> # Manuscript Revisions
>
> We have made the following changes to the manuscript:
>
> * Added a new experiment subsection, which describes the RLBench suite of tasks and presents our main results and includes some visualizations of the task.
> * Added the new sample efficiency ablation
> * Created a new original figure illustrating relative placement for the Problem Statement section.
>
> **Overall, we hope we have addressed the majority of the reviewers’ concerns about our experiments section. We believe that our revised manuscript is substantially stronger with the addition of RLBench experiments and analysis.**

---

### Comment · Area_Chair_Ub67 · 2023-11-21
**Reviewers: Please respond to authors or update review**

Dear Reviewers,

The discussion phase will end tomorrow.  Could you kindly respond to the authors rebuttal letting them know if they have addressed your concerns  and update your review as appropriate? Thank you.

-AC

---

### Author Response · Authors · 2023-11-22
**New experiments: 5 high-precision tasks on RLBench, our method outperforms baselines**

We have completed experiments on 5 new tasks from the [RLBench](https://sites.google.com/view/rlbench) suite, and believe we have produced strong evidence for the utility of our method on tasks with high precision requirements.

# Tasks:
- [Insert onto Square Peg](https://drive.google.com/file/d/1gzdkGmx-5Kd1AAgmv2LB9FkfhtM4vOWu/view?usp=sharing): A high-precision task involving taking a small ring and placing it on a peg (think Tower of Hanoi). There are only a few millimeters of clearance for this task.
- [Bottle on Rack](https://drive.google.com/file/d/1F3pLigfwgLJoe_-Ip_ikO4KKzOGfG_Lr/view?usp=sharing): A task for placing a wine bottle on a wine rack. Precision requirements are somewhat more tolerant, but still have only a few millimeters of tolerance.
- [Phone on Base](https://drive.google.com/file/d/19KWJqgg9eecFXg69Wvpx1e5V3wBZpRXG/view?usp=sharing): Place a home telephone on its base. Requires precision of a few millimeters.
- [Put Toilet Roll on Stand](https://drive.google.com/file/d/18IjsoKV3WFcQB8Dp2tFmXL0Dxl819en-/view?usp=sharing): place a roll of toilet paper onto a peg. Requires ~1cm of precision to succeed.
- [Place Hanger on Rack](https://drive.google.com/file/d/1c_jlMg9mMZr9HgGWbPWoVzi0FoTHq-yK/view?usp=drive_link): Place a thin clothes hanger on a thin clothes rod. ~1cm of precision required to succeed.

We will include screenshots of each task in the appendix of our revised manuscript.

# Experiment procedure:

1. Generate 10 demonstrations for each task using the provided expert agents for RLBench
2. Select the final image in the demonstration as the desired configuration for the object, and the first image as the initial configuration of the object.
3. Train a model to predict the rigid body transform which moves the object to the desired goal configuration. We use the proposed symmetry-breaking techniques from the original TAX-Pose paper to make the regression target unambiguous.
4. To evaluate, we sample 1000 unseen initial configurations of the objects in the scene, and predict the desired configuration of each object.
5. We compute both the average rotational error (in degrees) and the translation error (millimeters), which indicate how precisely the models have learned the desired relationship compared to ground-truth success demonstrations.

# Results Table:

|               |  stack_wine\\ angle_err (°) |   t_err (mm) |   put_toilet_roll_on_stand\\ angle_err (°) |   t_err (mm) |   place_hanger_on_rack\\ angle_err (°) |   t_err (mm) |   phone_on_base\\ angle_err (°) |   t_err (mm) |   insert_onto_square_peg\\ angle_err (°) |   t_err (mm) |
|:--------------|------------------------------:|--------------------------:|--------------------------------------------:|----------------------------------------:|----------------------------------------:|------------------------------------:|---------------------------------:|-----------------------------:|------------------------------------------:|--------------------------------------:|
| TAX-Pose       |                      1.47  |                3.09 |                                     1.17 |                              **1.25**   |                                5.47  |                          12.0  |                         4.14  |                   5.43 |                                   7.10  |                            3.52 |
| Ours (RelDist) |                      **0.76** |                **1.02** |                                     **1.15** |                              1.34 |                                **0.62** |                          **1.96** |                         **0.80** |                   **1.06** |                                   **1.21** |                            **3.29** |

We are in the process of preparing a properly-formatted version of this table for our manuscript.

Note: we do not report “overall task success rates” because of insufficient time in the rebuttal period to design and comprehensively evaluate a motion planning policy in the RLBench environment which performs both grasp and placement motions.

# Analysis

Across the board, our proposed representation RelDist leads to very high-precision task-specific pose predictions, with **less than 1.25 degrees in angular error and less than 2 millimeters of translational error on average compared to examples of successful placements**. TAX-Pose, however, frequently makes predictions with significantly higher angular and translational error. In high-precision tasks, rotational misalignment can easily cause task failure (for instance, with the peg insertion task), necessitating a high degree of rotational precision.

**We believe that this is strong evidence that our method is suitable for high-precision tasks, as it achieves very low error in absolute terms and significantly outperforms TAX-Pose on nearly every metric in every task.**

---

### Author Response · Authors · 2023-11-23
**New experiment: Strong sample efficiency for learning precise placement**

We have completed an additional ablation experiment to evaluate the sample efficiency of our method compared to TAX-Pose.

# Experiment Procedure

1. We select a single task from the RLBench set of tasks, specifically Stack Wine.
2. For each method, we train 3 different versions, where the training set consists of 1, 5, and 10 expert demonstrations of placements (subsets of our full dataset from the main experiments) for each version.
3. We then conduct the same evaluation as in the main results for the RLBench dataset, where we sample 1000 unseen initial placements of the object in the scene.
4. We report the relationship between the number of demonstrations and the angular / translation error on these unseen placements.

# Results Tables

## No. training demos vs. angular error (degrees)
**Task: Stack Wine**

|               |       1 demo |        5 demos |       10 demos|
|:--------------|--------:|---------:|---------:|
| TAX-Pose       | 4.26 | 5.00  | 1.27  |
| Ours | **1.04** | **0.73** | **0.75** |

## No. training demos vs. translation error (mm)
**Task: Stack Wine**

|               |         1 demo |          5 demos |         10 demos|
|:--------------|----------:|-----------:|-----------:|
| TAX-Pose       | 10.7 | 7.38 | 2.44 |
| Ours | **2.66**  | **1.42** | **1.21** |

# Analysis

Our method shows very high precision on this task when trained with only a single demonstration: approximately 1 degree of rotational error and <3mm of translation error. In contrast, TAX-Pose is substantially less precise given only a single demonstration. This trend holds across all three versions of the model. **Our method gains nearly all of its eventual precision after seeing only a single demonstration**. We believe that this behavior is a direct result of our provably-invariant encoder which needs to only learn a single invariant representation, as opposed to one which is equivariant. Furthermore, we observe in all cases that our network achieves to a highly-precise prediction (close to converged performance) in substantially fewer training steps than TAX-Pose, also due to the guaranteed invariance.

**We believe that this is strong evidence that our method is capable of learning highly precise relationships from even a single demonstration.**

---

### Meta-Review · Area_Chair_Ub67 · 2023-12-09

**Metareview:**

**Summary** This work presents a method for learning high precision relative poses in SE(3) for robotic manipulation tasks using demonstrations.  The problem is formatted in terms of finding a transformation SE(3) which will transform object A to a desired pose with respect to object B, for example moving a mug to hang on a mug tree. The method has two parts, the first part learns a task-specific relative distance matrix RelDist describing the desired location of object A with respect to object B specified in terms of distances between points in the point clouds representing the two objects.  This is learned using an SE(3)-invariant Vector Neurons based encoder, cross-attention, and a learned kernel.  The second part computes the relative transformation in SE(3) from RelDist using differentiable SVD.  The method is evaluated for precision on 5 RLBench task and for object generation on NDF relative placement tasks.  The method is also evaluated on offline data from real world sensors.

**Metareview**  This work proposes a novel method for handling high-precision robotic manipulation tasks in 3D.  The SE(3)-invariant ReLDist representation combined with multilateration is a clever way to incorporate problem symmetries.  The paper is well-written and clear.  The problem addressed is important and has practical applications.   Experimental evaluation generally shows the method to work well.  Results with respect to precision metrics on RLBench tasks (added during rebuttal) demonstrate the methods advantage in high precision tasks and also show high sample efficiency.  However, the method has some limitations in an inability to represent ambiguity due to object symmetry or multimodel placement task which hurt performance in some tasks such as bowl placement.  Also, the paper does not perform real-world experiments, which although not strictly necessary for demonstrating the value of the method here, would strengthen the paper.

**Justification For Why Not Higher Score:**

- limitations in representing ambiguity in pose due to object symmetry or multimodel placement task.  These hurt model performance in several experiments.
- no real world evaluation

**Justification For Why Not Lower Score:**

- novel and clever method
- empirical evaluation shows good precision and efficiency

---

### Decision · Program_Chairs · 2024-01-16

Accept (poster)